# Strengthening of the Somali upwelling during the Holocene and its impact on southwest monsoon rainfall

Balaji D.[1,2], Ravi Bhushan[2], L. S. Chamyal[1]

[1]Department of Geology, The Maharaja Sayajirao University of Baroda, India
[2]Geoscience Division, Physical Research Laboratory, Ahmedabad, India

*Correspondence to*: Balaji D. (balaji.d86@gmail.com)

**Abstract.** The history of the Somali upwelling during the last 18.5 ka has been reconstructed using biogenic silica fluxes estimated from a sediment core retrieved from the western Arabian Sea. Surface winds along the east African coast during southwest monsoon causes the Somali upwelling and therefore the intensity of this upwelling has been related to the southwest monsoonal variability. Variations in biogenic silica fluxes suggest periodic weakening and strengthening of the Somali upwelling. A weakened upwelling during the last glacial period (18.5-15 ka BP) and strengthened upwelling during the Bølling-Allerød (15-12.9 ka BP) suggest post-glacial onset of the southwest monsoon. Whereas the Younger Dryas (12.9-11.7 ka BP) is marked by reduced upwelling strength, intensification of the Somali upwelling at the beginning of the Holocene and a decline at 8 ka BP have been observed. Increase in upwelling strength recorded since 8 ka BP suggest strengthening of the southwest monsoon during the latter part of the Holocene. These upwelling variations when compared with the southwest monsoon precipitation record, a reversal in the relationship between the strength of the Somali upwelling and southwest monsoon rainfall is observed at the beginning of Holocene. The observed shift is attributed to the variation in the southwest monsoon strength due to the latitudinal shift of the Intertropical Convergence Zone (ITCZ) associated with changes in moisture sources.

## 1 Introduction

The greater part of the world's population inhabits the tropical region, where climate is mainly controlled by monsoon rainfall. Understanding the causes of past changes thus plays a critical role in deciphering past, present and future monsoon variability. The economy of India, which is a tropical country that contains a significant part of the world's population, is dependent to a large extent on the southwest monsoon (SWM) rainfall; hence, slight changes in SWM rainfall can lead to immense societal impacts. Several attempts have been made to identify the factors responsible for SWM rainfall variations. SWM rainfall variability is correlated with several global phenomena, such as ENSO (Goswami et al., 1999; Annamalai and Liu, 2005), Atlantic Sea Surface Temperature (SST; Goswami et al., 2006; Yadav, 2016), Eurasian snow cover (Hahn and Shukla, 1976; Pant and Rupa Kumar, 1997; Bamzai and Shukla, 1999), the pre-monsoon 500 hPa ridge (Mooley et al., 1986), the Indo-Pacific warm pool (Parthasarathy et al., 1988; Parthasarathy et al., 1991), the Pacific decadal oscillation (Krishnan and Sugi,

2003), and the Atlantic multi-decadal oscillation (Krishnamurthy and Krishnamurthy, 2015). In addition to these factors that influence SWM, the Indian Ocean Warm Pool (IOWP) has been identified as the prominent source of moisture for SWM rainfall (Ninomiya and Kobayashi, 1999; Gimeno et al., 2010). During its maxima, the IOWP extends throughout the northern Indian Ocean during the pre-monsoon period (April), and it almost reduces to half during the SWM (Izumo et al., 2008). The extent of the IOWP is primarily affected by the Somali upwelling and partly by the latent heat flux increase in the Arabian Sea

during the SWM season (Izumo et al., 2008). The Somali upwelling, as well as SWM rainfall, are caused by the SWM winds during boreal summer.

Upwelling of deep water during the SWM brings nutrients up to the photic zone, enhancing surface productivity in the western Arabian Sea. Paleoproductivity variations in the coastal regions off Somalia and Oman have been extensively studied to understand past changes in SWM-related upwelling (Sirocko et al., 1993; Naidu and Malmgren, 1996; Gupta et al., 2003;

Tiwari et al., 2010). However, variations in siliceous productivity in the western Arabian Sea, which have direct implications for the strength of upwelling in the past, have not been understood. The present study thus aims to understand past variations in siliceous productivity in the Somali upwelling region, as well as paleo-upwelling strength and its relationship with the southwest monsoon rainfall, using a sediment core retrieved from the western Arabian Sea (Fig.1).

### 1.1 Modern Oceanography and Productivity

The surface water circulation in western Arabian Sea is controlled by seasonal changes in atmospheric wind pattern related with annual migration of ITCZ (Wyrtki, 1973). During boreal winter, ITCZ stays south of equator and shifts to north during boreal summer. This northward shift of ITCZ during southwest monsoon (SWM, June-September) season drives the southern hemisphere eastern trade winds across equator that turns clockwise and becomes southwest winds (Findlater, 1977; Fig. 2). These southwest winds help to form the Somali current along the east African coast towards north. The Somali current is

generally associated with near shore upwelling and eddies such as southern gyre, great whirl and Soccotra eddy (Schott et al., 1990; Beal and Chereskin, 2003; Schott et al., 2009). These eddies induce intense upwelling which pumps out the low temperature and nutrient rich subsurface water to the surface along the east coast of Africa (Young and Kindle, 1994).

Productivity in the western Arabian Sea reflects the seasonal changes in surface ocean characteristics (Qasim, 1977; Brock et al., 1991). More than half of the annual productivity in the western Arabian Sea occurs during southwest monsoon due to

intense upwelling (Haake et al., 1993). Total flux (biogenic + dust) also peaks at the same time when productivity is at its maximum, indicating that SWM not only causes high productivity in the western Arabian Sea but also contributes to high dust flux (Sirocko and Lange, 1991; Haake et al., 1993). Bhushan et al., 2003 observed that the concentration of nitrate and phosphate increased at the bottom of the mixed layer at the core location, whereas, significant increase in silicate concentration occurs only at the depth of thermocline. During the onset of SWM upwelling, nitrate and phosphate rich waters surfaced more

compared to silicate due to the upwelling of shallow waters. In presence of high nitrate and phosphate along with the micronutrients (derived by dust flux), the calcareous primary producers dominated the surface productivity. Sediment trap studies in the western Arabian Sea recorded high biogenic carbonate flux at the onset of SWM upwelling (Haake et al., 1993).

During the late phase of SWM upwelling, the surfacing of deeper waters increased silicate concentration in surface waters. High silicate content and depletion of micro-nutrients to sustain excessive nutrient utilization results in siliceous productivity and subsequent biogenic silica flux to the sediments (Fig.3; Konning et al., 2001). Initially, this upwelled deep water surfaced at Somali coastal upwelling zone and transported towards the mouth of Gulf of Aden (core location) through the Socotra channel i.e. between the Socotra Island and Somalia (Young and Kindle, 1994).

## 2 Material and methods

Sediment core SS4018 was collected off the Horn of Africa (north of Socotra island), from the western Arabian Sea (13° 12.80' N, 53° 15.40' E; water depth 2830 m; core length 130 cm; Fig. 1) during FORV Sagar Sampada cruise SS-164 in 1998. Sub-sampling at 2-cm intervals was carried out. The age-depth model (Fig. 4) as well as the calcareous and organic productivity proxies of this sediment core have been presented elsewhere (Tiwari et al., 2010). Dry bulk density (DBD) was computed using an empirical equation based on the calcium carbonate concentrations (Clemens et al., 1987). The flux rate was estimated using an average sedimentation rate computed based on the age-depth model given by Tiwari et al., (2010). However, the published ages of the individual samples were considered. The sedimentation rate at the core site as given by Tiwari et al., (2010) is variable, the lowest being 3.5 and highest at 22.7 cm.ka-1. Since the age model depends on the sample selection criteria and may change according to depth of age control points, an average sedimentation rate was computed for the entire core.

The biogenic silica concentration was measured in each sample using the method described by Carter and Colman (1994). Dried homogenized samples weighing 50 mg were placed in centrifuge tubes. Five milliliters of 10 % $H_2O_2$ was added to each sample at room temperature, and the samples were stored for 2 hours to remove organic matter. Five milliliters of 1N HCl was added to each tube. After acid treatment, 20 ml of distilled water was added, and the samples were centrifuged for 15 minutes. Sample tubes were kept in an oven after removal of the supernatant. Thirty milliliters of 2 M $Na_2CO_3$ was added to each sample tube, and the tubes were kept in a shaker bath at $95^0$ C for 5 hours. After 5 hours, the samples were centrifuged for 5 minutes, and 3 ml of hot supernatant was pipetted out of each sample and added to exactly 30 ml of distilled water in pre-cleaned sample tubes. The solution was acidified by adding 0.9 ml of concentrated $HNO_3$. Sample tubes were sealed after effervescence. Silicon and Aluminium concentrations were measured in these samples using ICP-AES (Jobin-Yuvon, Model 38S at Physical Research Laboratory, Ahmedabad). The silicon concentrations were then corrected for clay mineral dissolution by using the formula given by Carter and Colman (1994) (Eqn 1):

$$\Delta Si = Si - (Al * 1.93) \tag{1}$$

Where, $\Delta Si$ is the corrected silicon concentration, Si and Al are the measured concentrations of silicon and aluminium in the sample, and 1.93 is the Si to Al ratio in smectite. Smectite is an abundant clay mineral in the northern Arabian Sea (Sirocko et al., 1991). Biogenic silica concentrations were calculated using the formula given below (Eqn 2):

$$Biogenic\ silica = \Delta Si * \mathcal{K} \tag{2}$$

Where, K is a constant that equals 2.4, which accounts for the ~10 % water content in biogenic silica (Mortlock and Frolich, 1989). Overall, the error associated with the biogenic silica measurement is less than 5 % based on repeat measurements. The biogenic silica flux is calculated by multiplying the biogenic silica fraction by the sedimentation rate (SR) and the dry bulk density (DBD) (Eqn 3),

$$B.Si.flux(g.m^{-2}.y^{-1}) = B.Si * SR(m.y^{-1}) * DBD(g.m^{-3}) \qquad (3)$$

The uncertainties associated with the biogenic silica concentration (B.Si) is estimated from the error in Aluminium and Silicon concentration based on measurements of repeat and standard material. The maximum error in biogenic silica concentration is within 5%. Dry bulk density (DBD) is calculated from CaCO3 concentration using an empirical equation suggested by Clemens et al., 1987. The standard uncertainty in DBD calculation is 0.091 g/cm3. The uncertainty in average sedimentation rate (SR) is 0.12 cm/ky. Finally the uncertainty associated with biogenic silica flux (B.Si flux) is propagated using the below

equation,

$$\sigma B.Si\,flux = B.Si\,flux * \sqrt{\{(\sigma B.Si/_{B.Si})^2 + (\sigma DBD/_{DBD})^2 + (\sigma SR/_{SR})^2\}} \qquad (4)$$

Where, prefix "σ" stands for uncertainty. Uncertainty in biogenic silica concentration is below 5%. But the uncertainty in flux are up to 15%. This increase in uncertainty is due to the high standard error associated with empirical derivation of Dry bulk density.

**3 Results**

    Biogenic silica concentrations varied from 3 % to 15.2 % during the last 18.5 ka (Table 1), with the lowest concentrations (3–4 %) being observed at the bottom of the core between 18.5 ka and 16 ka BP. Subsequently, it increased continuously up to 13.5 ka BP (~9 %) and decreased to 6.5 % at ~13 ka BP. No significant variations in biogenic silica concentrations were observed between 13 ka and 11.3 ka BP. After 11.3 ka BP, biogenic silica increased from 6.5 % to 10.5 % and remained stable

for a period of almost 1000 years until 10 ka BP. The biogenic silica concentration increased from 10.5 % to 12 % during 10–9.5 ka BP, and it decreased subsequently to 8.5 % at 8 ka BP. From 8 ka to 7.3 ka BP, it changed from 8.5 % to 9.5 % with a peak at 7.7 ka BP (13.3 %), further it remained stable until 6 ka BP. From 6 ka to 5 ka BP, the biogenic silica concentration increased from 10 % to 12.5 % and then decreased at 4 ka BP. After 4 ka BP, a continuous increase to a maximum value of 15 % biogenic silica at 1.5 ka BP and a subsequent decrease to 12 % at 1 ka BP were observed. Biogenic silica concentrations

showed no variations during the last 1 ka BP.

    The variations in biogenic silica fluxes show an overall increasing trend from 18.5 ka to present (Fig.5). The fluxes varied between 1.4 to 6.8 g.m$^{-2}$.y$^{-1}$. The minimum flux (<2 g.m$^{-2}$.y$^{-1}$) was observed at the bottom of the sediment core, i.e., between 18.5 and 16 ka BP, and the maximum flux at 2 ka BP. Five distinct peaks in biogenic silica flux during the last 18.5 ka BP were observed between 14–13 ka, 11–10 ka, 10–9 ka, 6–4 ka and 2.5–1 ka BP. Uncertainties in biogenic silica concentrations

and fluxes are below 5 % and up to 15 %, respectively. The increase in uncertainty in the flux is due to the high standard error associated with the empirical derivation of the dry bulk densities.

## 4 Discussion

### 4.1 Biogenic silica flux as an upwelling proxy

The surface waters of the world ocean are mostly deficient in bioavailable silica (Hurd, 1973), which is a major nutrient for
siliceous productivity. Apart from the Southern Ocean, high siliceous productivity can be observed in the major upwelling
regions, where upwelled nutrient-rich water causes high primary production (Koning et al., 2001). The ocean is under saturated
with respect to silica, and thus biogenic silica flux in sediments is a function of its export flux, which is controlled by its
production at the surface and dissolution in the water column as well at the sediment water interface (Hurd, 1989; Broecker
and Peng, 1982). Thus, using biogenic silica as a proxy for the study of paleo-upwelling requires understanding of its
production and burial efficiency. Sediment trap studies from the western Arabian Sea indicated biogenic silica flux mimics the
SWM upwelling (Fig. 3; Nair, 2000; Haake et al., 1993). Studies of sediment trap data and surface sediments (Koning et al.,
1997; Koning et al., 2001) of the Somali basin provides better estimates of the burial efficiency of biogenic silica i.e. ratio
between diatom abundance in surface to its concentration in the sediment, in the western Arabian Sea. Only 6.8–8.7 % of
diatom (biogenic silica) productivity is preserved in the sediments of the Somali basin; the rest is remineralised in the water
column and at the sediment water interface (Koning et al., 2001). One of the major findings from sediment traps in the Somali
basin by Koning et al., (2001) is the selective preservation of upwelling-indicating diatoms in the sediments of the Somali
region. This is linked to the silicification of diatom frustules; most pre- and post-upwelling produced diatoms are weakly
silicified, enhancing their dissolution in the water column and leading to their low preservation in sediments. Better
preservation of upwelling indicating diatoms may also be linked to the increased downward supply due to high surface
production. Nutrient availability (Si:N) and concentration of dissolved iron can also affect diatom silicification that leads to
variation in preservation (Hutchins and Bruland, 1998). In general, it is noted that high silicate concentration along with micro-
nutrient depletion leads to more silicified and faster sinking diatoms (Hutchins and Bruland, 1998), which is most plausible
scenario during the late phase of SWM upwelling. If burial efficiency (BE) is the primary controller of biogenic silica flux
variation, ratio of low flux to low BE should have been similar to the ratio of high flux to high BE. However, in the present
record using the modern high and low BE values (Koning et al., 2001), the ratio of high flux to high BE (top) is almost three
times more than the ratio of low flux to low BE (bottom), thereby indicating the absence of preservation effect in the biogenic
silica flux.

Apart from biogenic silica production and preservation efficiency, sediment redistribution can also influence the biogenic silica
flux. However, considering the sediment core location and average sedimentation rate, it is likely that the influence of sediment
focusing/winnowing on the flux record is minimal. The location of the sediment core is far from the continental slope (Fig. 1)
and not directly influenced by coastal currents or fluvial systems that would lead to redistribution of the sediment flux. The
chronology of the sediment record is based on the published age-model given by Tiwari et al., (2010). The model, however,
shows sizeable variations in the sedimentation rate. Because the calculated sedimentation rate is a function of the selected
sampling depths (which may vary), average rate of sedimentation has been considered in estimating the fluxes of biogenic

silica. The high surface production of biogenic silica during SWM upwelling (Fig. 3; Haake et al., 1993; Koning et al., 1997), together with the increased burial efficiency of upwelling-indicating diatoms (biogenic silica) in the western Arabian Sea sediments (Koning et al., 2001), makes biogenic silica flux as a potential proxy for SWM-related upwelling in the study area.

## 4.2 Biogenic silica flux vs SST

Similar to biogenic silica flux, the paleo-SST reconstructions can serve as proxy for upwelling, as upwelling increases siliceous

productivity with reduction in SST. However, the inverse relation between siliceous productivity and SST is valid only during the SWM season and not even on annual scale. Mostly the SST reconstructions are based on biomarker ($TEX_{86}$ and $U^{K'}_{37}$; Brassell et al., 1986; Schouten et al., 2002) or from planktonic foraminifera shell chemistry ($\delta^{18}O$ and Mg/Ca; Emiliani, 1955; Chave, 1954). However, the annual mean signal of SST or its reconstruction on seasonal scale depends on the signature of the productivity proxies for the region. The $\delta^{18}O$ value of foraminifera shell is not only dependent on the SST, but also gets

modulated with the preservation of shell, carbonate ion concentration, salinity and $\delta^{18}O$ (ice-volume) of the original seawater (Lea, 2003), which makes the reconstruction rather complex. The SST reconstruction based on Mg/Ca ratio is affected by species-dependency and dissolution (Lea, 2003). In most of the cases, it is foraminifera based SST which preserves better signature of annual mean signal due to the foraminifera production throughout the year (Conan et al., 2000; Dahl and Oppo, 2006), but the possibility of preserving seasonal signal depends on local hydrography and productivity. While, the biomarker

based SST records may preserve seasonal signal, because they are mainly produced during monsoon season (Huguet at al., 2006). However, there are other limitations with the biomarker based SST records. Major limitation for the application of alkenone $U^{K'}_{37}$ in low latitude regions is that it saturates around 28∘C (Prahl and Wakeham, 1987). While, the $TEX_{86}$ based SST are valid in the range of 5 to 30 C (Kim et al., 2008), with better estimates above 15ºC (Kim et al., 2010). However, there are numerous evidence suggesting $TEX_{86}$ records sub-surface temperature rather than SST (Hertzberg et al., 2016; Lee et al.,

2008; Lopesdos Santos et al., 2010; Seki et al., 2012; Wuchter et al., 2006). The nutrient availability and variation in productivity may also influence the $TEX_{86}$ temperature (Hertzberg et al., 2016). Comparing the SST reconstructed using $U^{K'}_{37}$ and $TEX_{86}$ from a sediment core in the western Arabian Sea, Huguet et al., 2006 suggested that the $U^{K'}_{37}$ SST are in phase with northern hemisphere dynamics during NE monsoon, while, TEX86 SST are controlled by SW monsoon. All SST proxies tend to record annual mean signal with varying fraction of seasonal signal. Glacial boundary conditions have strong influence

on the annual mean SST in the Arabian Sea irrespective of monsoon upwelling (Broccoli, 2000; Dahl and Oppo, 2006). While, the biogenic silica flux has been controlled by the SWM upwelling signal due to the fact that it is produced during southwest monsoon season and preserves upwelling signal. Hence, the biogenic silica flux can be identified as a better proxy than SST to understand SWM upwelling in the study area. Comparison of biogenic silica flux with other paleo-SST records (Anand et al., 2008; Sahar et al., 2007; Huguet et al., 2006) from nearby locations are shown in figure 6 and 8. There is no definitive

relation between biogenic silica flux and SST records on temporal scale. Also, the SST using different proxies show inconsistent changes in the studied time span, the $TEX_{86}$ SST is always higher than Mg/Ca SST (Fig. 6&8). A general observation is that both biogenic silica flux and SST were low during 18.5 to 15 ka BP, later showing an anti-correlation (Fig.

6). The anti-correlation is strong between biogenic silica flux and $TEX_{86}$ SST during 15 to 11.7 ka BP (Fig. 6). However, during the last 11.7 ka the Mg/Ca based SST shows a strong anti-correlation with biogenic silica flux record, indicating variation in influence of the seasonal signal for different proxies (Fig. 8). Temporal relation between biogenic silica flux and SST is discussed in the following section.

### 4.3 Somali upwelling strength versus southwest monsoon rainfall

The western Arabian Sea SSTs during SWM are directly related to upwelling strength (enhanced upwelling results in lower SSTs and vice versa; Fig. 3). Previous studies have shown that northern Indian ocean and in particular the Arabian Sea to be an important source of moisture for SWM rainfall over India (Ninomiya and Kobayashi, 1999; Gimeno et al., 2010). There are several kind of relationship observed between SST, moisture and SWM rainfall. First order relation would be positive, i.e. reduced SWM winds would cause reduced upwelling as well as reduced rainfall and vice versa. However, the relation between Arabian Sea SST and SWM rainfall is complicated due to the fact that SST modulates the moisture availability as well as the meridional temperature gradient (Levine and Turner, 2012). Modelling study by Shukla (1975) showed that the cold Arabian Sea SST during SWM tend to reduce the SWM rainfall through reduced moisture transport. However, Webster et al (1999) and Clark et al (2000) showed that the SWM rainfall has stronger connection with winter and spring SSTs rather than summer, and suggested a delayed influence of SST on rainfall. A modelling study by Arpe et al (1998) demonstrated that warmer northern Indian Ocean leads to increased SWM rainfall over India through enhanced evaporation and moisture supply, while indicating the strong influence of pacific SST anomalies on monsoon. It was also suggested that Arabian Sea SST modulates the impact of ENSO on monsoon precipitation (Arpe et al., 1998; Lavine and Turner, 2012). An observational study by Vecchi and Harrison (2004) detected a strong positive correlation between western Arabian SSTs and SWM rainfall over the Western Ghats Mountains in India from 1982 to 2001. Overall, it has been suggested that any isolated cooling of the Arabian Sea will reduce SWM rainfall through reduced moisture supply, in absence of other large-scale forcing (Lavine and Turner, 2012). An observational and modelling study by Izumo et al (2008) signals the causes for the variations in western Arabian Sea SSTs and their influence on SWM rainfall over the Western Ghats. According to Izumo et al (2008), increased Somali upwelling during the late spring reduces the westward extension of the IOWP during summer, which decreases moisture availability to the air mass that delivers rainfall to the western part of the Indian sub-continent. Though, the upwelling-rainfall connection is not fully understood and difficult to model, the observations suggest an anti-correlation between Somali upwelling (western Arabian Sea SST during SWM) and SWM rainfall. Both Somali upwelling and SWM rainfall were caused by southwest monsoonal winds during SWM season, hence their anti-correlation indicates a negative feed-back within the system.

Did this anti-correlation exist between the Somali upwelling and SWM rainfall in the geological past? To answer this question, we need to investigate the record of paleo-upwelling in the Somali region and paleo-rainfall in the western part of India and adjoining areas. There is no continuous terrestrial record of paleo-rainfall covering the last 18.5 ka from the Western Ghats, but there are several paleoclimatic records based on marine sediment cores from the eastern Arabian Sea. The biogenic silica flux temporal variability is compared (Fig. 7 & 10) with paleo-rainfall record ($\delta^{18}O_w$ IVF by Anand et al., 2008)) from

the eastern Arabian Sea and a speleothem record from Oman (Fleitmann et al., 2003). The $\delta^{18}O_w$ IVF is the Ice Volume Free oxygen isotopic composition of seawater based on the $\delta^{18}O$ of *G.ruber*. Anand et al. (2008) showed that the reconstructed $\delta^{18}O_w$ IVF during the last 19 ka from a sediment core (Sk-17) in the eastern Arabian Sea was mainly controlled by the SWM rainfall in the Western Ghats. The Qunf speleothem record from Oman (Fleitmann et al., 2003) had been widely used as an indicator for SWM variation. The location of Qunf speleothem is very close to the present study area i.e. downwind side to the present study area during SWM season. If the SWM was the reason for the rainfall in southern Oman, then western Arabian Sea must be the moisture source. Though, there are no observational study on the relation between upwelling strength and rainfall in Oman, a comparison is made to give a preliminary assessment. Since records are from different regions and have irregular temporal resolution, only long-term trends have been examined.

### 4.3.1    Last Glacial Period (18.5–15 ka BP)

The biogenic silica flux record does not show any distinct variation between the previously identified Heinrich event 1 and the Last Glacial Maximum (LGM; Clark et al., 2009), the period between 18.5 and 15 ka thus is considered here as the Last Glacial Period (LGP). Both biogenic silica concentrations (3–5 %) and fluxes (~2 g.m-2.y-1) were lowest during the LGP (Fig. 5), similar to earlier findings of low productivity during glacial periods from the western Arabian Sea (Burckle, 1989; Sirocko et al., 1991; Sirocko et al., 2000; Ivanochko et al., 2005; Tiwari et al., 2010). Based on the modern pattern of biogenic silica productivity and its burial efficiency in the western Arabian Sea, the observed low fluxes of biogenic silica indicate that the Somali upwelling was very weak during the LGP. However, the lowest SSTs recorded in the last 18.5 ka in the Somali basin during the LGP (Huguet et al., 2006; Saher et al., 2007; Anand et al., 2008) are related to basin wide cooling and not connected with upwelling strength (Fig. 6). Dahl and Oppo, 2006 observed a reduction in Arabian Sea SST of 2–4° C during LGP. Thus, it is unlikely that the IOWP (SST>28° C) formed during LGP. In absence of IOWP, any relation between the Somali upwelling and rainfall is unexpected. Paleo-rainfall record from the eastern Arabian Sea shows high $\delta^{18}O_w$ IVF values indicative of reduced freshwater flux and rainfall during this period (Fig. 7). Based on weak upwelling in the western Arabian Sea and reduced fresh water influx to the eastern Arabian Sea, it can be concluded that the SWM was weak/absent during the LGP.

### 4.3.2 Deglacial Period (15–11.7 ka BP)

The Deglacial Period (DP) is a connecting phase between two entirely different climatic periods, the LGP and the Holocene. The DP is basically a composite of two millennial scale events between 15–12.9 ka and 12.9–11.7 ka BP. This period occupied by these two events nearly coincides with well-known climatic events, specifically the Bølling-Allerød (B/A) and the Younger Dryas (YD) event. The beginning of the B/A is marked by an abrupt increase in biogenic silica flux (Fig. 6a) is attributed to the effect of northern limit of southwest monsoon, attained at the study site with subsequent increase in Somali upwelling strength. This is further supported by Zr/Hf in two independent sediment cores near our core site, which shows an increasing flux of windborne dust from the Horn of Africa (an indicator of the SWM) at the onset of the B/A (Sirocko et al., 2000; Isaji

et al., 2015). The reduction in TEX$_{86}$ SST during the B/A in the Somali basin (Huguet et al., 2006) also suggest increased upwelling (Fig. 5), however the Mg/Ca SST does not show this change (Fig. 6). The inconsistency between the two SST records (TEX$_{86}$ and Mg/Ca) could be related to the control on seasonal production of the proxy material.

The $\delta^{18}O_w$ IVF record from core SK-17 (Anand et al., 2008) shows depleted values, indicating higher influx of fresh water from the Western Ghats caused by high SWM rainfall, during the B/A (Fig. 7c). The positive correlation between Somali upwelling (high biogenic silica flux) and SWM rainfall in the Western Ghats (high fresh water influx to the eastern Arabian Sea) during the B/A contrasts with the present-day scenario as observed by Vecchi and Harrison (2004). Presently, the moisture source for SWM rainfall is the Arabian Sea and the central Indian Ocean (IOWP), which is affected by SWM upwelling (Izumo

et al., 2008). If the central Indian Ocean were the source of moisture for SWM rainfall during the B/A, then the observed co-variation is possible. Thus, it is proposed that the moisture source for SWM rainfall over Western Ghats during the B/A event was different from the modern source. The other possibility, that rainfall in south-western India was enhanced due to a strong NE monsoon during the B/A is unlikely because the siliceous productivity in the western Arabian Sea related to the NE monsoon has not been reported (Koning et al., 1997; Ramaswamy and Gaye, 2006). In contrast to the B/A, the upwelling in

the western Arabian Sea was weak during the YD, as revealed by the low biogenic silica fluxes and high SSTs (Fig. 6; Huguet et al., 2006). This is in agreement with the previous studies from Arabian Sea which showed decreased productivity during YD due to reduction in SWM (Altabet et al., 2002; Ivanochko et al., 2005). Furthermore, the high $\delta^{18}O_w$ IVF in the eastern Arabian Sea (Anand et al., 2008) were caused by low freshwater influx points to weak SWM rainfall (Fig.7).

### 4.3.3 Holocene (11.7–0 ka BP)

The beginning of the Holocene is marked by an abrupt increase in biogenic silica flux (Fig. 8a). This sudden increase in biogenic silica fluxes between 11.7 ka and 9 ka BP could have been caused by the intensification of the SWM (extended season) resulting in northward shift of the ITCZ, following the peak in Northern Hemisphere solar insolation (Fleitmann et al., 2007). Somali basin SST records (Saher et al., 2007; Huguet et al., 2006; Anand et al., 2008) also shows a marginal decrease at the onset of the Holocene, but not up to the levels seen during the B/A (Fig.8). The TEX$_{86}$ SST show more variation than

Mg/Ca at the beginning of Holocene, however, the Mg/Ca SST shows a mirror image to the biogenic silica flux pattern during Holocene, indicates dominance of seasonal signal in Mg/Ca SST during this period (Fig. 8b). Based on the stable isotopic composition of organic carbon and nitrogen in the 4018 core, Tiwari et al (2010) too suggested an increase in productivity during Holocene and attributed it to the strengthening of Somali upwelling. The synchronous changes in the biogenic silica flux with biogenic silica/carbonate ratio (Fig. 9) indicates a change in dominant plankton community (carbonaceous to

siliceous) due to increased upwelling, as suggested by Tiwari et al. (2010). The $\delta^{18}O_w$ IVF (Anand et al., 2008) display values like those of the YD during the early Holocene (Fig. 10c), indicating reduced rainfall (lower fresh water influx) over the Western Ghats. This anti-correlation between Somali upwelling and SWM rainfall over south-western India during the early Holocene (11.7 ka to 9 ka BP), marks the establishment of the modern-day climate system. The increased Somali upwelling in the western Arabian Sea during the early Holocene (11.7 to 8 ka BP; Fig. 10a) might have reduced the IOWP expanse during

the SWM season, thereby resulting in lower moisture availability and subsequent reduced rainfall over the Western Ghats. Oman speleothem record also shows decreased precipitation during early Holocene (Fig. 10b) and supports this interpretation of low moisture availability.

At ~8 ka BP (Fig. 10a), Somali upwelling strength decreased compared to the early Holocene, but persisted above YD and B/A levels, indicating the presence of the SWM with reduced wind strengths relative to the early Holocene. The SST record

also shows an increased value at ~8 ka BP (Fig. 8b) indicating reduction in Somali upwelling. This reduction in upwelling at 8 ka BP can support the westward extension of the IOWP during the SWM season, thereby increasing both moisture availability over the Arabian Sea and rainfall over the Western Ghats. The $\delta^{18}O_w$ IVF value decreased at 8 ka BP, indicating an increase in fresh water influx from the Western Ghats (Anand et al., 2008) due to increased SWM rainfall (Fig. 10c). Oman speleothem record too shows decreased $\delta^{18}O$ value at 8 ka BP (Fig. 10b) suggesting increased SWM rainfall.

Somali upwelling had a gradual increase during the last 8 ka with minor positive excursion at around 5 and 2 ka BP (Fig. 10a). The increase in SWM induced Somali upwelling during the last 8 ka contrasts the idea that SWM followed the northern hemisphere insolation during Holocene (Gupta et al., 2003; Fleitmann et al., 2003 and references therein). However, this interpretation is well in agreement with other studies from the Arabian Sea which shows that SWM did slightly increase during Holocene (Agnihotri et al., 2003; Tiwari et al., 2010). These short-term increase in Somali upwelling at 5 and 2 ka BP can also

be observed with reduction in the Mg/Ca SST record (Fig. 8b). Oman speleothem record shows an increase in $\delta^{18}O$ during the last 8 ka suggesting reduction in SWM rainfall (Fig. 10b). The hiatus in Oman speleothem record at 2 ka BP coincides with the strengthened Somali upwelling, however, it is difficult to explain (Fig. 10a & b). The SK-17 record (Anand et al., 2008) shows slight increase in the $\delta^{18}O_w$ IVF of surface waters during the last 8 ka (Fig. 10c) indicating reduction in SWM rainfall. The opposite trend in upwelling and rainfall record during the last 8 ka indicates the negative impact of Somali upwelling on

SWM rainfall through changing the area of IOWP and moisture availability. However, the short term variations in upwelling is not observed in the eastern Arabian Sea rainfall record.

The Somali upwelling possibly had a negative impact on southwest monsoon rainfall over south-western India and Oman throughout the Holocene. This finding would have implications in context of the modelling study by deCastro et al. (2016), which shows that Somali upwelling would increase during the twenty-first century.

**5 Conclusions**

The present study demonstrates the use of biogenic silica flux as a proxy for the temporal variations in the strength of the Somali upwelling during the last 18.5 ka. Some of the salient findings of the present study are summarized below:

1. The Somali upwelling was weak during the LGP coeval with the weak southwest monsoon.
2. The post-glacial onset of the southwest monsoon was marked by an increase in the strength of the Somali upwelling
at 15 ka BP, with the eastern Arabian Sea records showing increased southwest monsoon rainfall.

3.  The Somali upwelling was weak between 12.9 and 11.7 ka BP, indicating another phase of weak southwest monsoon similar to that of the LGP. Overall, records of the Somali upwelling and southwest monsoon rainfall exhibit positive correlations between 18.5 and 11.7 ka BP.

4.  Shift in positive to negative correlation between the strength of the Somali upwelling and southwest monsoon rainfall occurred at 11.7 ka BP at the beginning of the Holocene, which marks the establishment of modern day climate system.

5.  Enhanced Somali upwelling during the last 11.7 ka BP, except for the decline at 8 ka BP, had a negative impact on southwest monsoon rainfall.

6.  Both, latitudinal shifts in the Intertropical Convergence Zone (ITCZ) and changes in moisture source region act as causative factor for the reversal in the relationship between upwelling and southwest monsoon rainfall.

7.  Future observational and modelling studies on southwest monsoon rainfall reconstruction and prediction should incorporate variations in the moisture source region.

**Acknowledgments**

Balaji D. is thankful to the Council of Scientific and Industrial Research (CSIR) of India for providing support through a CSIR-NET-Senior Research Fellowship. Ravi Bhushan thanks the Director of the PRL for research grant support. We thank Dr. Alpa Sridhar for critical comments and suggestions on the manuscript. We thank the two anonymous reviewers for their constructive comments which helped us in improving the manuscript.

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

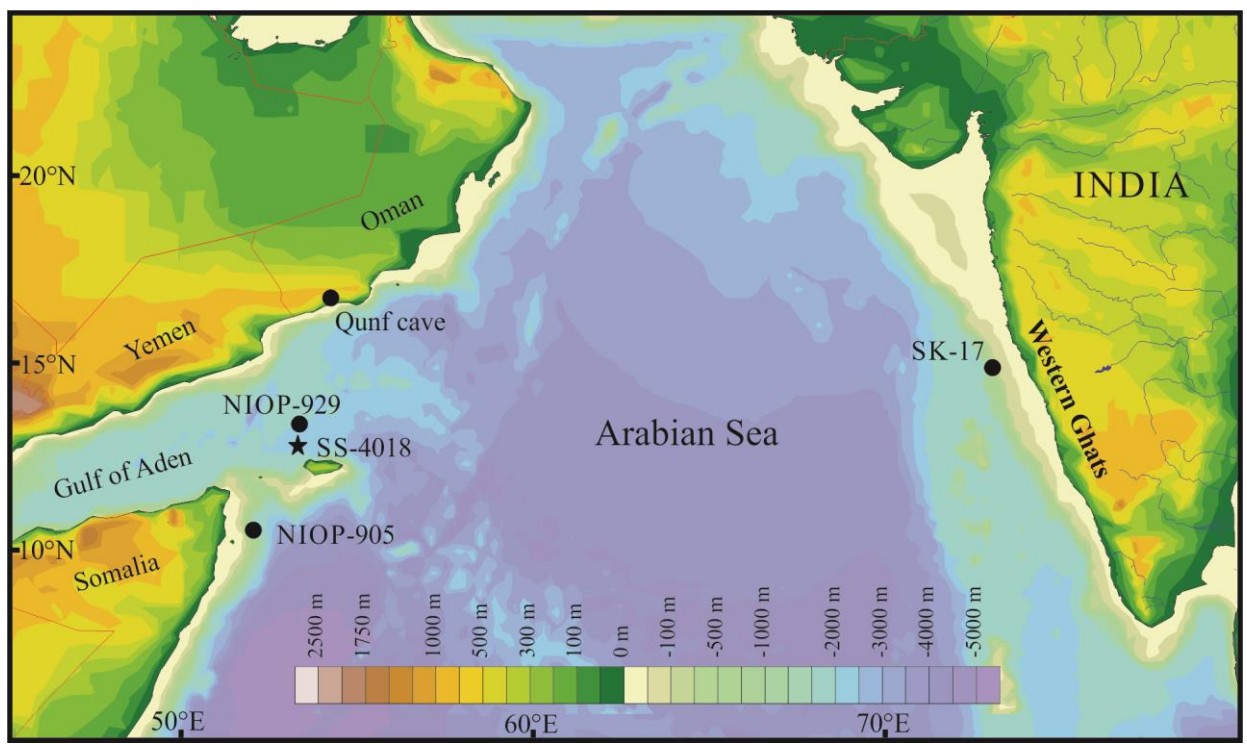

**Figure 1: Location of sediment core SS-4018 (filled star) in the Arabian Sea. Also shown are the sites discussed in the paper: NIOP-929 (Saher et al., 2007), NIOP-905 (Huguet et al., 2006), SK-17 (Anand et al., 2008), Qunf cave (Fleitmann et al., 2007).**

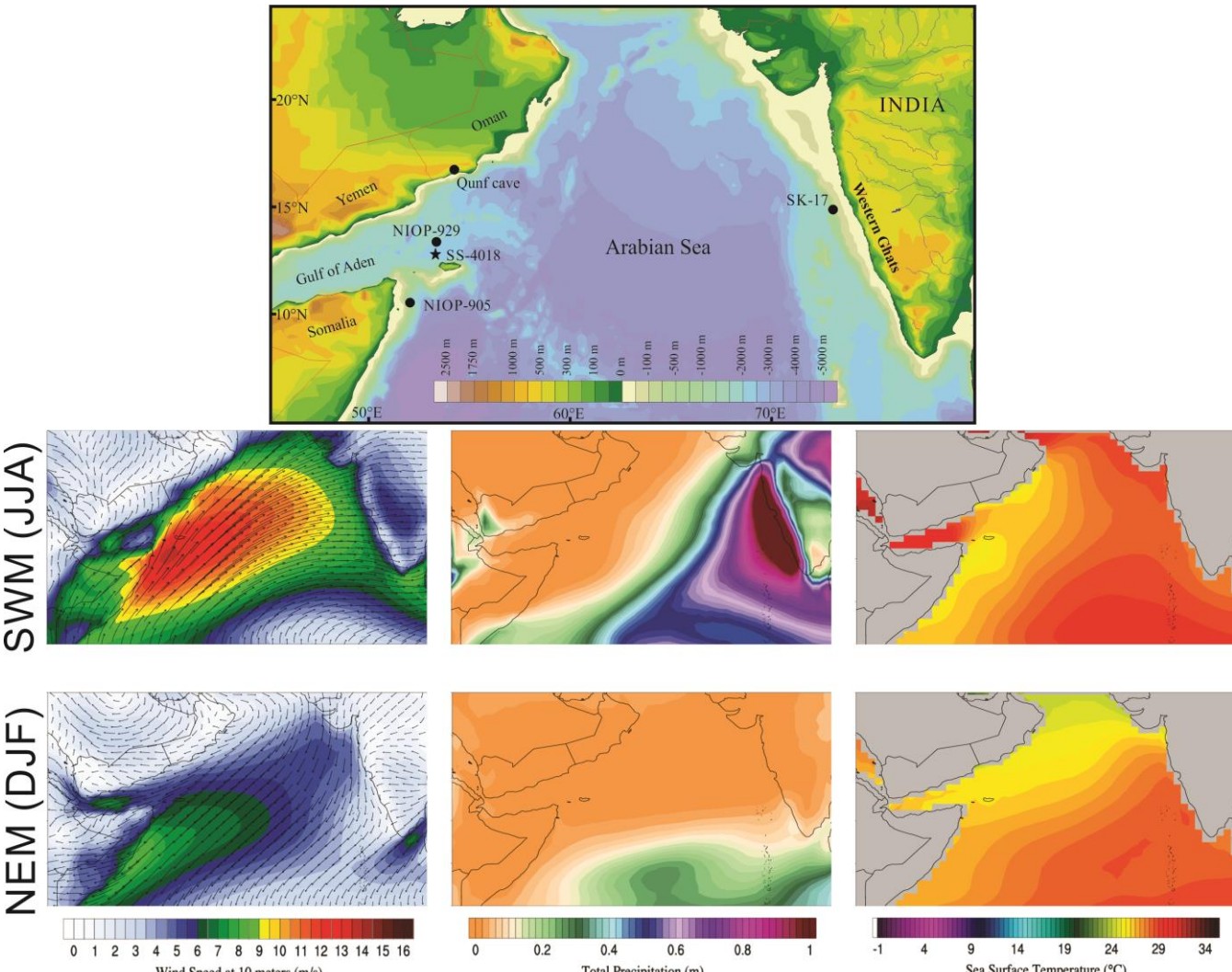

Figure 2: Location of our sediment core SS4018 in the western Arabian Sea. Also shown are the sites discussed in the manuscript. Bottom figures shows the seasonal changes in Wind speed, Precipitation and SST during southwest (SWM) and northeast monsoon (NEM). ECMWF-ERA-Interim data (Berrisford et al., 2011) used and the image obtained using Climate Reanalyzer (http://cci-reanalyzer.org), Climate Change Institute, University of Maine, USA.

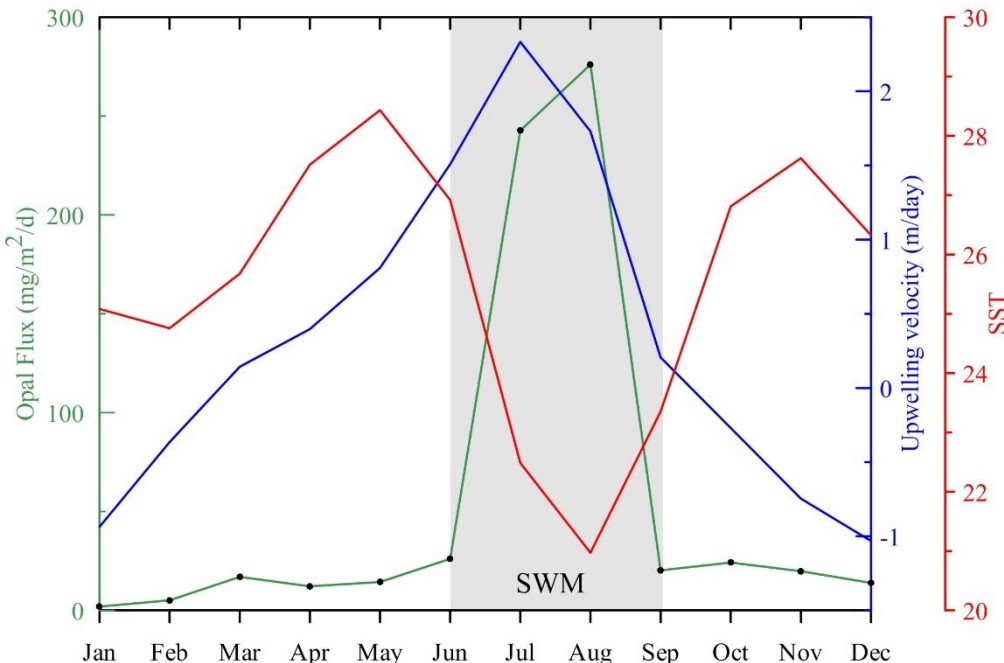


**Figure 3. Modern oceanography of western Arabian Sea. Synchronous change in upwelling intensity and biogenic silica flux clearly indicate that the siliceous productivity in western Arabian Sea is controlled by SWM upwelling. Upwelling strength data is used from Nair, 2000 and Opal flux from Haake et al., 2003.**

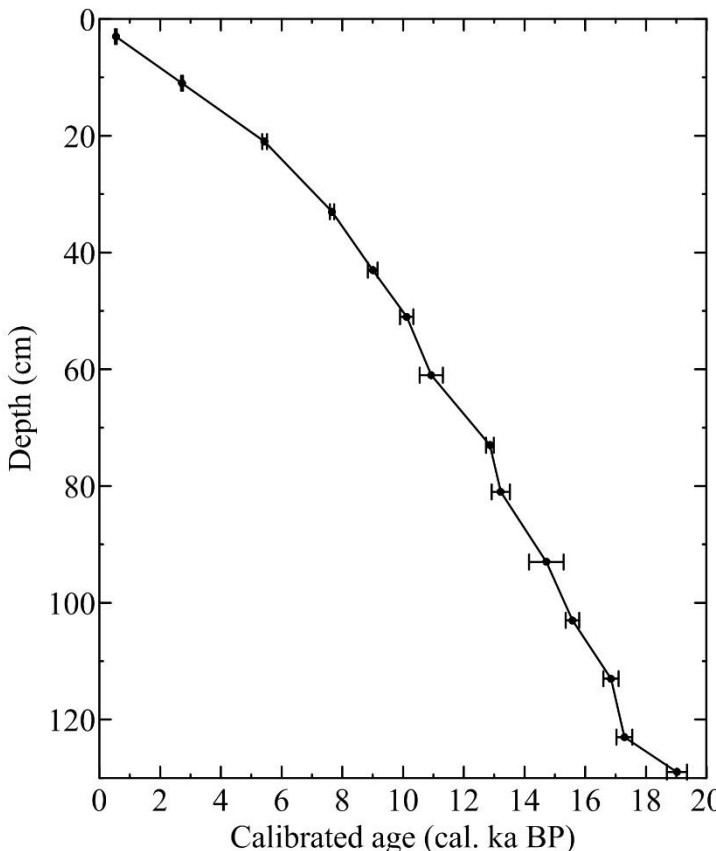

**Figure 4: Age depth model of the sediment core SS-4018 (adopted from Tiwari et al., 2010). The error bars marks one sigma uncertainty in calibrated age.**

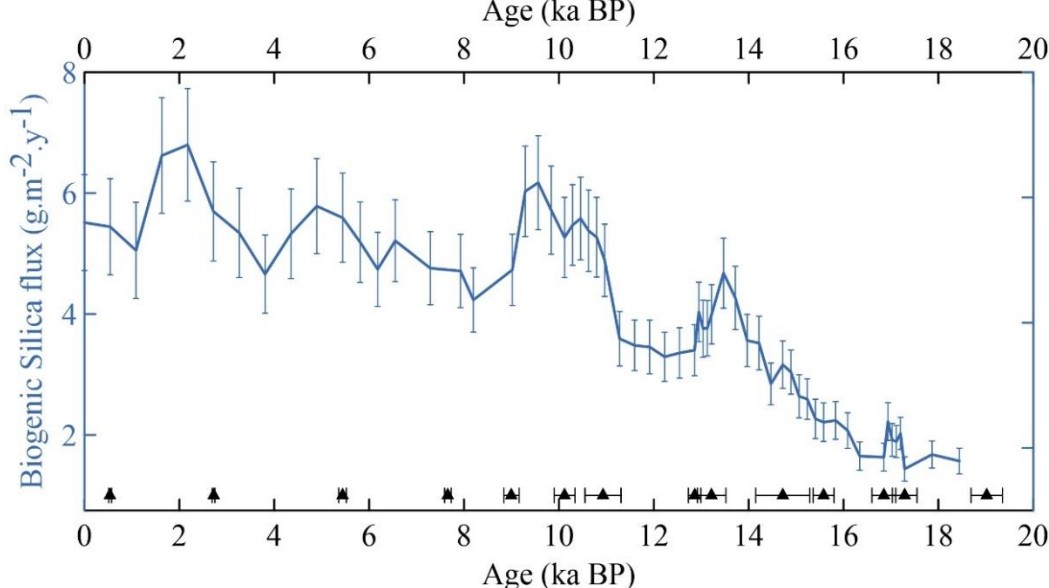

**Figure 5: Temporal variation of Biogenic silica flux with two sigma uncertainty in sediment core SS4018. Filled triangles at the bottom of the plot marks the age-control points with one sigma uncertainty.**

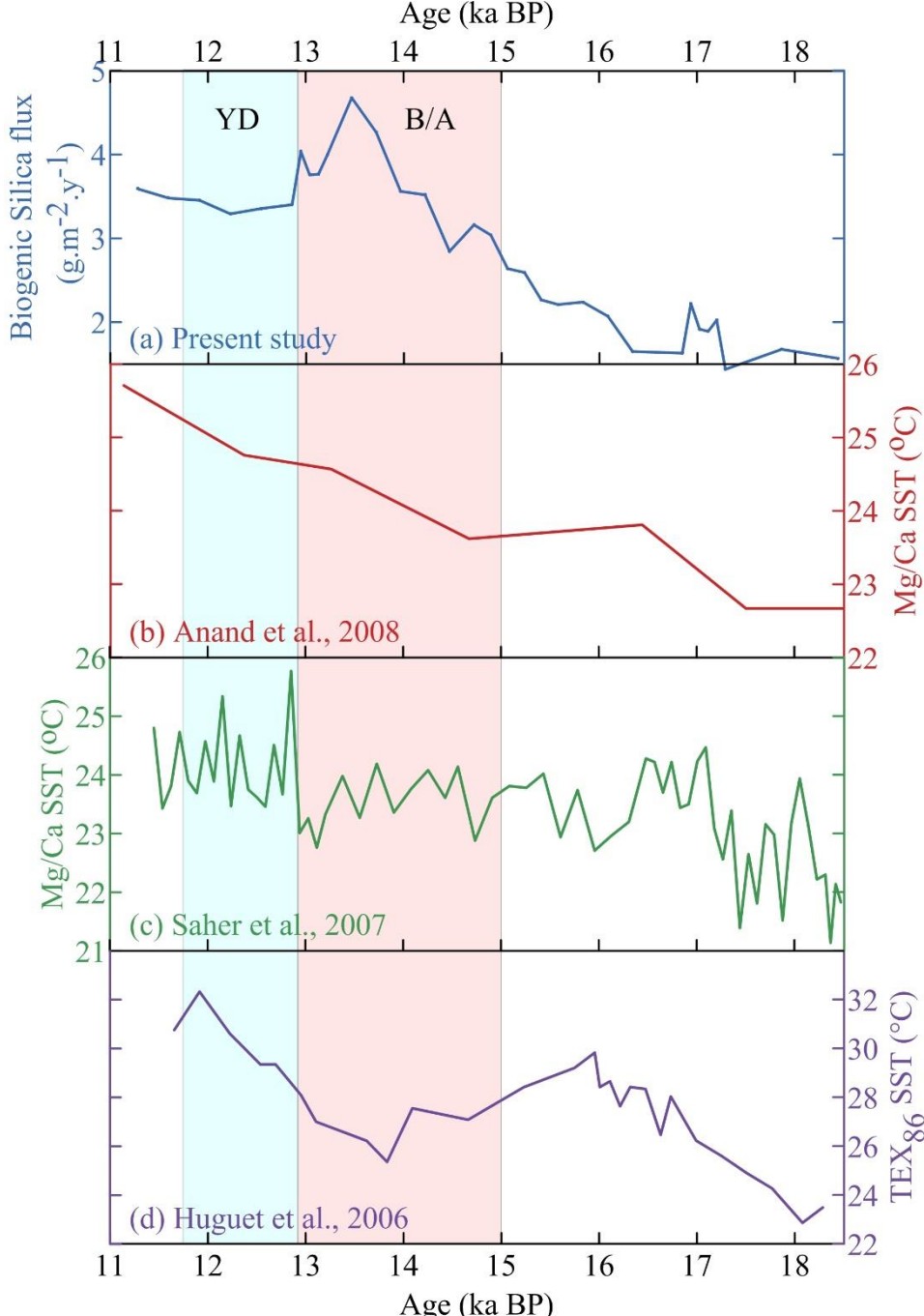

**Figure 6. Comparison of biogenic silica flux with SST records from western Arabian Sea for pre-Holocene time (18.5-11.7 ka BP). a)Biogenic silica flux, b) Mg/Ca based SST from NIOP-905 core (Anand et al., 2008), c) Mg/Ca based SST from NIOP-929 core (Saher et al., 2007), d) TEX$_{86}$ SST from NIOP-905 core (Huguet et al., 2006).**

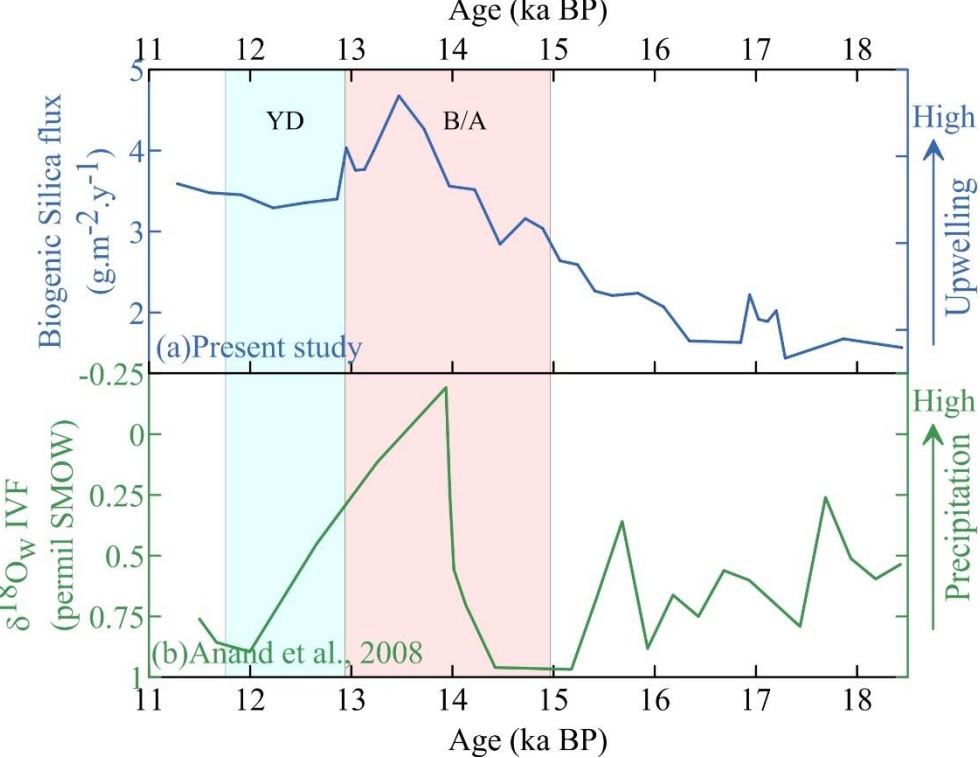


**Figure 7. Comparison of biogenic silica flux with rainfall record from eastern Arabian Sea for pre-Holocene time (18.5-11.7 ka BP). (a) Biogenic silica flux (present study), (b) $\delta^{18}O_w$ IVF (Anand et al., 2008).**

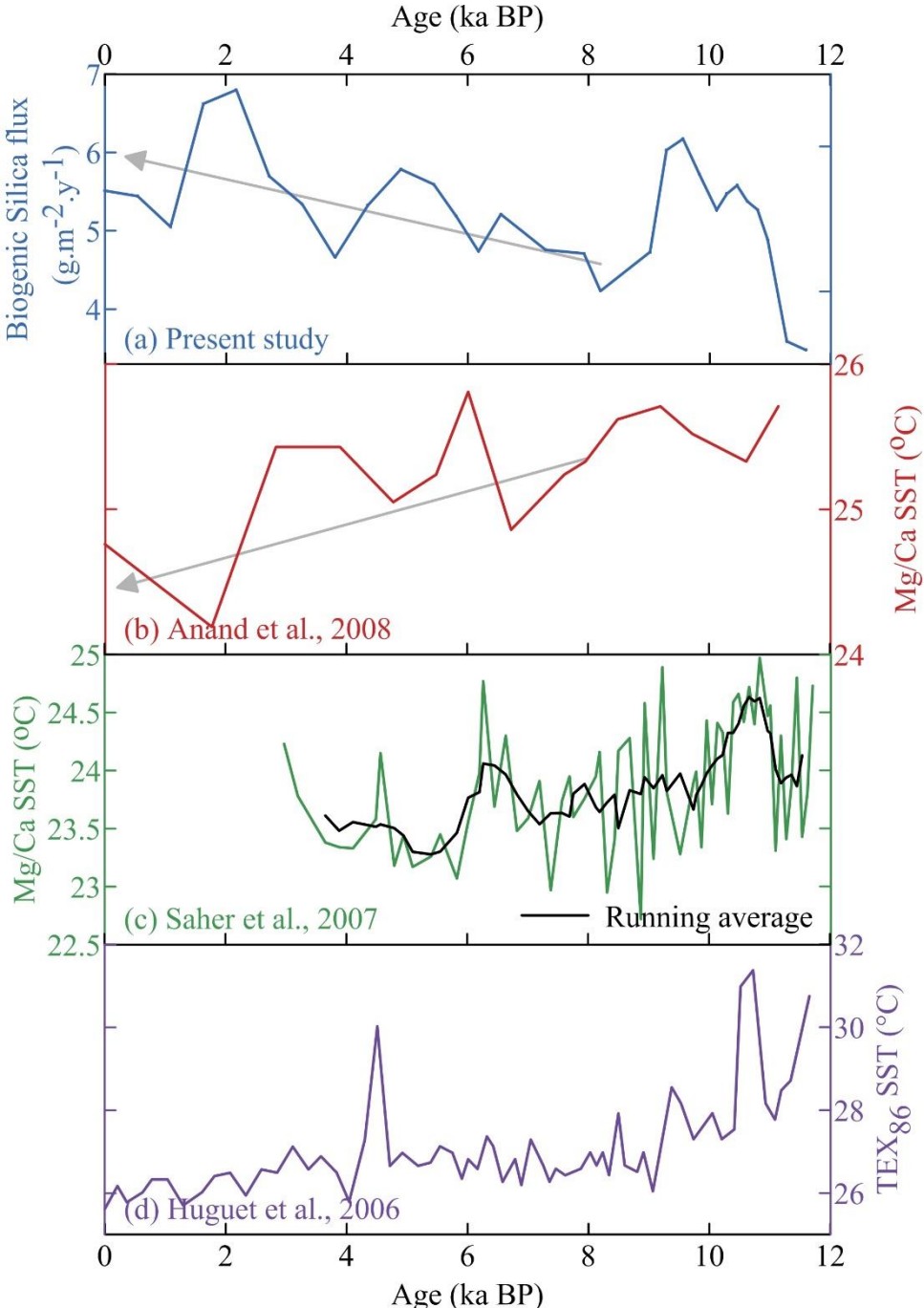

**Figure 8: Comparison of Somali upwelling with western Arabian Sea SST records. a) Biogenic silica flux, b) Mg/Ca based SST from NIOP-905 core (Anand et al., 2008), c) Mg/Ca based SST from NIOP-929 core (Saher et al., 2007), d) TEX86 SST from NIOP-905 core (Huguet et al., 2006). Grey arrow indicate the trend of proxy records during the last 8 ka.**

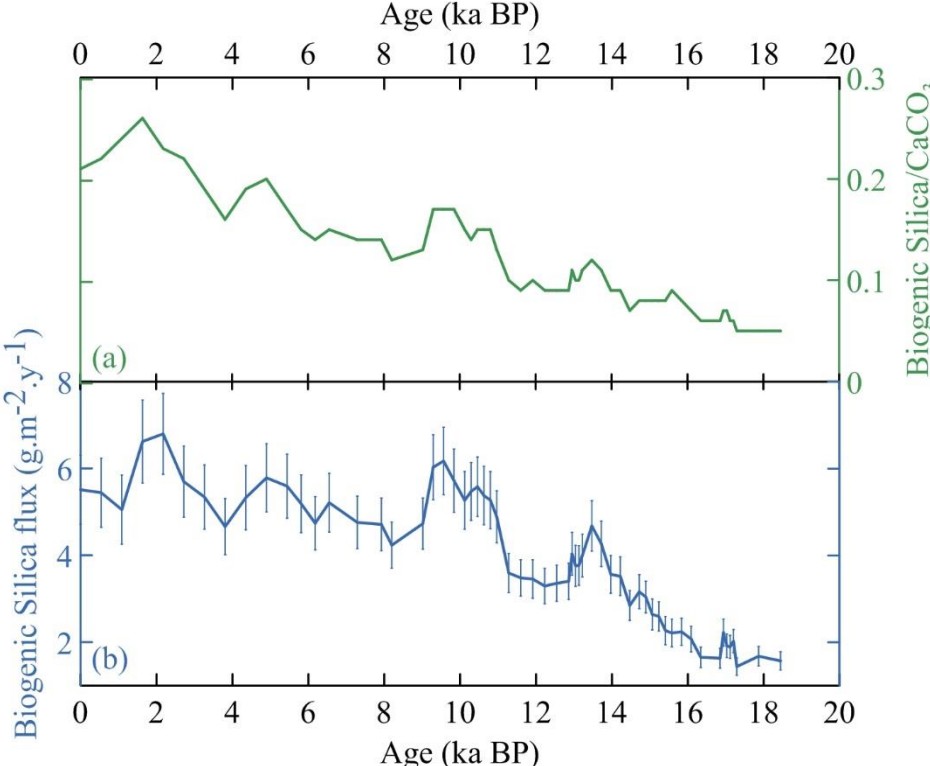

**Figure 9: Comparison of biogenic silica flux with silica to carbonate ratio in 4018 sediment core. Synchronous changes in both parameters indicate the dominance of biogenic silica flux on the ratio.**

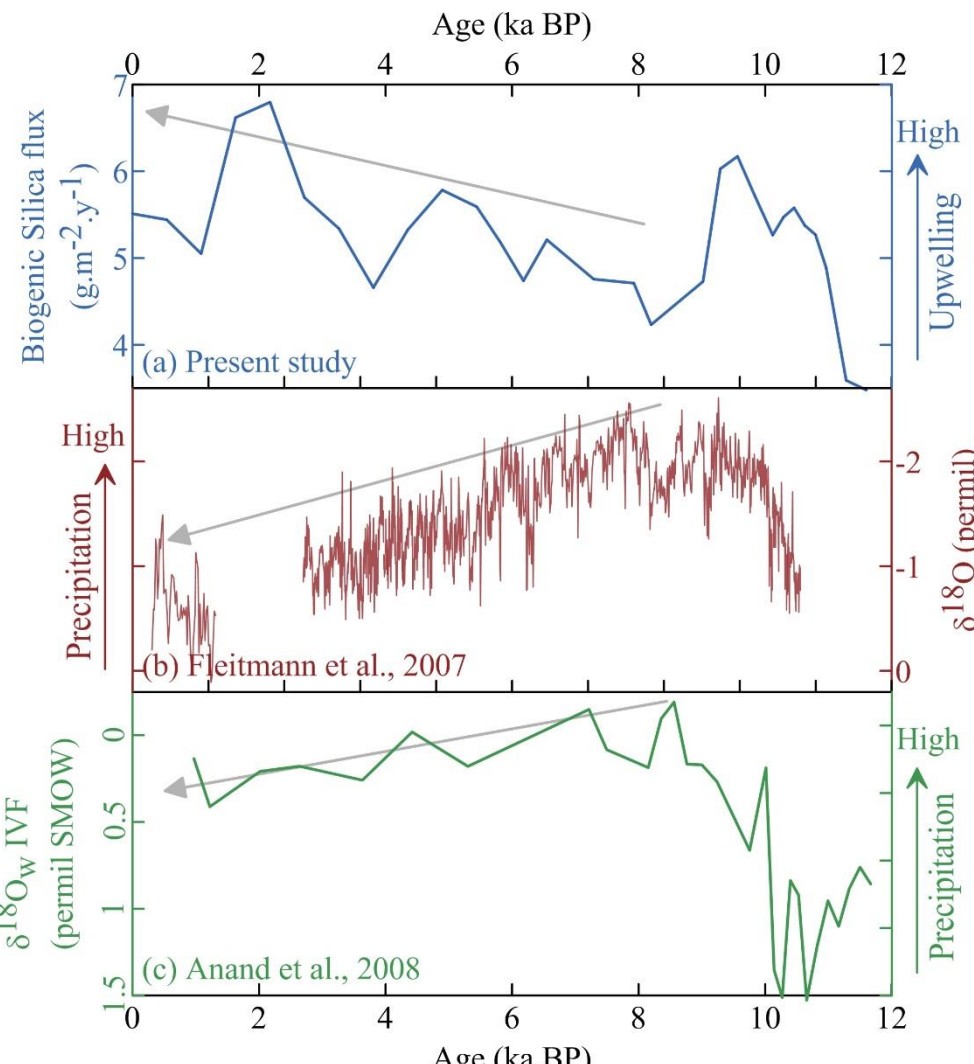

**Figure 10: Comparison of Somali upwelling with southwest monsoon rainfall records during Holocene. (a) Biogenic silica flux (present study), (b) Oman speleothem record, (c) $\delta^{18}O_w$ IVF data from eastern Arabian Sea. Grey arrow indicate the trend of proxy**
**record during the last 8 ka.**


| Age (ky) | B. Si (%) | σ B.Si | B.Si flux (g/m²/y) | σ B.Si flux | Age (ky) | B. Si (%) | σ B.Si | B.Si flux (g/m²/y) | σ B.Si flux |
|---|---|---|---|---|---|---|---|---|---|
| 0.00 | 12.58 | 0.22 | 5.51 | 0.79 | 12.23 | 6.41 | 0.11 | 3.29 | 0.41 |
| 0.54 | 12.58 | 0.25 | 5.44 | 0.80 | 12.54 | 6.49 | 0.16 | 3.36 | 0.42 |
| 1.09 | 12.58 | 0.28 | 5.05 | 0.80 | 12.86 | 6.49 | 0.20 | 3.40 | 0.42 |
| 1.63 | 15.20 | 0.20 | 6.62 | 0.96 | 12.95 | 7.76 | 0.13 | 4.03 | 0.49 |
| 2.17 | 14.81 | 0.14 | 6.80 | 0.93 | 13.04 | 7.37 | 0.18 | 3.76 | 0.47 |
| 2.72 | 13.06 | 0.05 | 5.70 | 0.82 | 13.13 | 7.24 | 0.09 | 3.77 | 0.46 |
| 3.26 | 11.70 | 0.21 | 5.34 | 0.74 | 13.22 | 7.74 | 0.13 | 3.99 | 0.49 |
| 3.81 | 10.23 | 0.17 | 4.66 | 0.65 | 13.47 | 9.06 | 0.21 | 4.68 | 0.58 |
| 4.35 | 11.79 | 0.10 | 5.33 | 0.74 | 13.72 | 8.23 | 0.16 | 4.27 | 0.52 |
| 4.90 | 12.49 | 0.13 | 5.78 | 0.79 | 13.97 | 6.83 | 0.11 | 3.56 | 0.43 |
| 5.44 | 11.64 | 0.22 | 5.59 | 0.74 | 14.22 | 6.83 | 0.22 | 3.52 | 0.44 |
| 5.81 | 10.52 | 0.17 | 5.19 | 0.67 | 14.47 | 5.41 | 0.10 | 2.84 | 0.34 |
| 6.18 | 9.74 | 0.04 | 4.74 | 0.61 | 14.72 | 6.11 | 0.16 | 3.16 | 0.39 |
| 6.55 | 10.52 | 0.29 | 5.21 | 0.68 | 14.89 | 5.74 | 0.09 | 3.04 | 0.36 |
| 7.29 | 9.59 | 0.12 | 4.76 | 0.61 | 15.06 | 5.49 | 0.19 | 2.64 | 0.36 |
| 7.93 | 9.59 | 0.15 | 4.71 | 0.61 | 15.24 | 5.24 | 0.14 | 2.59 | 0.34 |
| 8.20 | 8.31 | 0.19 | 4.23 | 0.53 | 15.41 | 5.07 | 0.15 | 2.27 | 0.32 |
| 9.02 | 9.35 | 0.10 | 4.73 | 0.59 | 15.58 | 5.08 | 0.04 | 2.21 | 0.32 |
| 9.29 | 11.88 | 0.09 | 6.03 | 0.75 | 15.83 | 4.94 | 0.08 | 2.24 | 0.31 |
| 9.56 | 12.28 | 0.19 | 6.17 | 0.78 | 16.09 | 4.53 | 0.17 | 2.07 | 0.29 |
| 9.84 | 11.57 | 0.14 | 5.72 | 0.73 | 16.34 | 3.66 | 0.11 | 1.65 | 0.23 |
| 10.12 | 10.43 | 0.21 | 5.26 | 0.66 | 16.85 | 3.54 | 0.13 | 1.63 | 0.23 |
| 10.29 | 10.57 | 0.10 | 5.47 | 0.67 | 16.94 | 4.75 | 0.20 | 2.22 | 0.31 |
| 10.46 | 10.82 | 0.14 | 5.58 | 0.68 | 17.03 | 4.17 | 0.15 | 1.92 | 0.27 |
| 10.63 | 10.59 | 0.20 | 5.38 | 0.67 | 17.11 | 4.04 | 0.17 | 1.89 | 0.27 |
| 10.80 | 10.45 | 0.11 | 5.27 | 0.66 | 17.20 | 4.17 | 0.08 | 2.02 | 0.26 |
| 10.97 | 9.47 | 0.12 | 4.89 | 0.60 | 17.29 | 3.05 | 0.12 | 1.44 | 0.20 |
| 11.28 | 7.04 | 0.17 | 3.59 | 0.45 | 17.87 | 3.52 | 0.09 | 1.68 | 0.22 |
| 11.60 | 6.59 | 0.09 | 3.48 | 0.42 | 18.44 | 3.25 | 0.11 | 1.57 | 0.21 |
| 11.91 | 6.92 | 0.17 | 3.45 | 0.44 | | | | | |

Table 1: Biogenic silica concentration and flux data.