# Peer review of "Strengthening of the Somali upwelling during the Holocene and its impact on southwest monsoon rainfall"

_Climate of the Past, 2017_

## Referee Comment (RC1) · Anonymous Referee #1 · 29 Sep 2017

In this paper, Balaji et al., use the biogenic silica flux from the Somali upwelling area to reconstruct regional climate over the last 18.5 ka. I found the data/results relatively straight forward but the results were slightly over-interpreted in conjunction with other datasets. I think part of the issue might be resolved in annotating the figures with specific intervals better (e.g. YD and BA).

One of my major concerns is using TEX86 values as sea surface temperature in an upwelling region. TEX86 values record variable temperatures in upwelling regions (see Hertzberg et al., EPSL 2016). TEX86 essentially records subsurface conditions in upwelling regions so interpreting the Huguet et al., 2006 record purely as sea surface

temperature is incorrect. I would recommend excluding this temperature record from the regional climate discussion. This would change the discussion section significantly. In particular, much of the YD and BA discussion hinges on the TEX86 record.

For comparisons for specific events like the YD and BA, I have some concerns about the age models. Many of the regional records are relatively low resolution. Are the age models, and frankly the sampling resolution, adequate the make regional interpretations for the YD and BA? I think annotating these specific intervals on the figures would help the reader and the authors evaluate these interpretations.

Could the authors clarify the selective preservation within the upwelling region (Lines 125-135)? In addition to the upwelling diatoms being more heavily silicified, presumably the flux of diatoms to the sediment during upwelling when nutrients are abundant also enhances their preservation? I'm not certain how the authors determined the preservation efficiency in the core. Is this based on the types of diatoms within the sediment?

Minor comments:

Please refer the reader to the specific figure (or figures) within the text. Figures should be labeled with region (e.g. Western Ghats) in addition to the author.

Line 15: "positive to negative" is ambigulous. Could you make this sentence a little more clear? Line 25: SST not defined.. write out sea surface temperature Lines 170-175: I think it would be best to start this line of argument with the colder SSTs in the LGP are related to global cooling and not a change in upwelling strength Line 175: use suggested instead of envisaged Line 188: I don't see a reduction in temperatures during the B/A Lines 209-201: I don't see this pronounced SST decrease in the Mg/Ca SST records, this is a TEX signal? Lines 231-235: I would recommend omitting this impact statement

---

## Author Comment (AC1) · 16 Oct 2017

**Reply to the Referee #1 comments**

The authors are thankful to referee for his/her critical and constructive comments which helped in improvising the manuscript. The necessary changes in view of the comments made by the referee has been incorporated in the manuscript and would be send along with those as would be suggested by other reviewers. Here, we provide replies to the comments.

**Comment:** *I found the data/results relatively straight forward but the results were slightly over-interpreted in conjunction with other datasets. I think part of the issue might be resolved in annotating the figures with specific intervals better (e.g. YD and BA).*

***Reply:*** The biogenic silica flux has been primarily used as proxy to reconstruct the Somali upwelling with SST records as secondary/supporting proxy. The data has been compared with the long-term trend of available palaeoclimatic records and have been summarized to the most appropriate conditions prevailed in the region. As suggested, the B/A and YD intervals have been marked in the figures 5 and 6.

**Comment:** *One of my major concerns is using TEX86 values as sea surface temperature in an upwelling region. TEX86 values record variable temperatures in upwelling regions (see Hertzberg et al., EPSL 2016). TEX86 essentially records subsurface conditions in upwelling regions so interpreting the Huguet et al., 2006 record purely as sea surface temperature is incorrect. I would recommend excluding this temperature record from the regional climate discussion. This would change the discussion section significantly. In particular, much of the YD and BA discussion hinges on the TEX86 record.*

***Reply:*** We agree that the $TEX^{86}$ proxy records surface or sub-subsurface sea surface temperature on longer time scales dependant on the nutrient availability and surface productivity. The suggested publication Hertzberg et al., EPSL 2016 from east pacific region indicates the $TEX^{86}$ record co-vary with Mg/Ca SST during modern and Holocene, but differs during the LGM. However, our aim is to decipher the changes in SST with respect to the biogenic silica variation as recorded in the present study. Since we are not discerning quantitative changes in SST or the inconsistencies between various SST proxies, only the relative trend of increase / decrease in SST has been considered. Moreover, as seen in Fig.5, the Mg/Ca based SST record by Saher et al., 2007 also shows co-variance with minor amplitude. As suggested, we have modified the discussion accordingly.

**Comment:** *For comparisons of specific events like the YD and BA, I have some concerns about the age models. Many of the regional records are relatively low resolution. Are the age models, and frankly the sampling resolution, adequate to make the regional interpretations for the YD and BA? I think annotating these specific intervals on the figures would help the reader and the authors evaluate these interpretations.*

*Reply:* The sampling resolutions of the studies discussed were sufficient enough to make comparison and interpret multi-millennial scale events like B/A & YD. Although, the age models of the discussed sediment cores are of low resolution, but reasonably high for events like B/A and YD, thus we have not attempted to compute any correlation plot between regional records. The B/A and YD intervals have been marked in the figures 5 and 6 as per referee's suggestion.

**Comment:** *Could the authors clarify the selective preservation within the upwelling region (Lines 125-135)? In addition to the upwelling diatoms being more heavily silicified, presumably the flux of diatoms to the sediment during upwelling when nutrients are abundant also enhances their preservation? I'm not certain how the authors determined the preservation efficiency in the core. Is this based on the types of diatoms within the sediment?*

*Reply:* For determining the preservation efficiency of diatom in the core, we have used findings of previous study by Konning et al., 1997 and 2001 from the Somali basin, wherein preservation efficiency is based on sediment traps and surface sediment studies. They observed "two well-silicified upwelling species, T. *Nitzschioides* and the solution-resistant *Chaetoceros*, make up ~60% of the sediment, and dominate sediments both in the core tops and down core, thereby preserving a residual upwelling signal". The estimated preservation efficiency is derived from the productivity and type of diatoms preserved in sediments (Konning et al., 1997 and 2001). Detailed discussion has been incorporated in the manuscript.

**Minor Comments:**

**Comment:** *Please refer the reader to the specific figure (or figures) within the text. Figures should be labeled with region (e.g. Western Ghats) in addition to the author.*

*Reply:* The suggested corrections have been incorporated in the text as well as in the figure captions of the revised manuscript.

**Comment:** *Line 15: "positive to negative" is ambiguous. Could you make this sentence a little more clear?*

*Reply:* The suggested change has been made.

**Comment:** *Line 25: SST not defined write out sea surface temperature*

**Reply:** Sea surface temperature is introduced at line 25 of the revised manuscript.

**Comment:** *Lines 170-175: I think it would be best to start this line of argument with the colder SSTs in the LGP are related to global cooling and not a change in upwelling strength.*

**Reply:** The statement has been modified as suggested.

**Comment:** *Line 175: use suggested instead of envisaged*

**Reply:** The change has been made.

**Comment:** *Line 188: I don't see a reduction in temperatures during the B/A*

**Reply:** The B/A event marked between 15-13 ka BP shows prominent reduction in temperatures both by Anand et al, 2008 and Huguet et al., 2006, though, Mg/Ca SST by Saher et al., 2007 shows minor change comparatively. This is noticeable with B/A events marked in the comparison figures.

**Comment:** *Lines 209-211: I don't see this pronounced SST decrease in the Mg/Ca SST records, this is a TEX signal?*

**Reply:** Thanks for the observation as there is no pronounced SST decrease in the Mg/Ca SST records, however, Mg/Ca SST record of Saher et al., 2007 indicates only a marginal decrease at the beginning of Holocene period. The amplitude in the SST change differs between Mg/Ca and TEX[86] records, with latter showing prominent change. Necessary modification made in the text.

**Comment:** *Lines 231-235: I would recommend omitting this impact statement.*

**Reply:** This statement has been made to highlight the significance of the present study. However, the statement being emphatic in nature has been toned down and appropriately modified as "The Somali upwelling can possibly have a negative impact on southwest monsoon rainfall over south-western India throughout the Holocene. This finding would have implications in context of the modeling study by deCastro et al. (2016), which shows that Somali upwelling would increase during the twenty-first century."

---

## Referee Comment (RC2) · Anonymous Referee #2 · 18 Dec 2017

The paper by Balaji et al. presents an interesting data set of biogenic opal measurements from a core from the Somali upwelling and discusses the data with respect to upwelling intensity off Somalia in relation with precipitation over the Western Ghats on peninsular India.

The authors compare their biogenic opal record, as a productivity/upwelling record, with salinity records from the eastern Arabian Sea which probably reflect precipitation in the Western Ghats. The authors also use the rainfall record from the Qunf cave (Fleitmann et al., 2007) on the Arabian Peninsula. Recent observational data and the modelling data presented by Izuma et al. (2008) indicate that upwelling off Oman

and Somalia and moisture transport to India are anti-correlated affecting rainfall in the Western Ghats and thus runoff to the eastern Arabian Sea. Sea surface salinities could be recorded in (residual) delta18O records of calcifying plankton of published core records SK 19 and AAS9/21 from the eastern Arabian Sea. The authors also compare their biogenic opal record with SST reconstructions from the Somali upwelling (NIOP 905 and 929).

The age model and other productivity proxies of the same core have been published by Tiwari et al. (2010). The new interpretation with respect to indicators of precipitation needs more discussion of available literature on monsoon climate.

I have three general remarks and suggestions. If addressed adequately this would lead to complete rewriting of the paper and also require some new interpretation of the data.

- The comparison of the productivity record with the SST records is misleading. As the authors state, high productivity could be expected during periods of strong up-welling, i.e. low temperatures. The SST records, however, are dominated by the strong glacial-interglacial temperature increase. So during this phase it looks like SST and productivities are positively correlated. The authors discuss this (chapter 4.2.1.) and thus start the comparison with a phase when it does not work; so the Figure still does not help much with the data interpretation. A comparison of other productivity records, concentrating on the Somali and Oman upwelling areas could be more illustrative. The SST records of the Holocene (after the strong glacial-interglacial) may be plotted with reversed scale in order to better illustrate whether and when there is an anti-correlation. In addition: there is some discussion in Huguet et al. (2006) about the TEX86 temperatures; it may not represent annual average temperatures but has a SW monsoon bias. This needs to be addressed and could actually support the authors.

- The comparison of the biogenic opal fluxes form the Somali upwelling and the delta18O (precipitation) records is a new idea (not published by Tiwari et al.) but is

too vague to be the main part of the paper. The anti-correlation of Western Ghats precipitation with western Arabian Sea upwelling was modelled for the present Arabian Sea and the authors cite only one paper (Izuma, see above). As the authors also discuss, differences in evaporation and also surface water inflow from the Bay of Bengal have impacted the salinity off the west coast of India during the past so that much of the changes are related to several different processes (see Vijit et al., 2016; Mahesh and Banakar, 2014). Furthermore, even the present relationship between precipitation on the Indian Subcontinent and upwelling/productivity in the Arabian Sea is not very clear (see Levine and Turner, 2012) so this topic needs at least some further discussion.

- In the paper by Tiwari et al. (2010), which the authors cite, more data on core SS4018 are available such as carbonate contents and stable isotopes of carbon and nitrogen. These data can be utilized to better understand the processes in the Somali upwelling and would help to better understand the Holocene productivity changes. Tiwari et al. come to similar conclusions, e.g. that productivity does not decline during the late Holocene despite the decreasing insolation, based on a multiproxy study. The authors have now additional evidence that this is the case and can prove what Tiwari et al. suggested: the decline of carbonate could be due to the replacement of carbonaceous by siliceous primary producers. The carbonate/opal ratio could show this and strengthen the authors' point. The published data need to be included and elaborated on.

Throughout the text there are many questions arising which need clarification and more detailed discussion:

The authors use the term "glacial" and "deglaciation" without giving references for these phases. They should also give the correct time for the beginning of the Holocene. I think that the use of LGP is rather uncommon but LGM is more common and can be referenced (Clark et al, 2009).

Lines 55-59: it does not become clear why biogenic silica appears after carbonate, clarify in more detail.

Line 68: Is the age model used here different from the one used by Tiwari et al. for the same core, if yes, why? Is the same rate of sedimentation used for the whole core, despite available C14 ages? Why?

Lines 130-134: difficult to understand, explain in more detail. Why should variations be three times greater?

Lines 145-148: very short and therefore difficult to understand, explain in more detail (see also general comment on the comparison of SST and productivity records above). When do you expect a correlation, when an anti-correlation, why? This cannot be explained in two sentences.

Line 181: I find the use of the deglacial period (DP; 15-11 ka BP) rather problematic as it covers the Pleistocene/Holocene boundary.

Line 185: what does "entrainment of the SW monsoon" mean?

Lines 197-204: these lines again show that the comparison of moisture and upwelling does not work (see above). So when does it work and is it at all useful to show it for the whole period?

Lines 221: very short and not convincing. How does the record from the Qunf Cave come into the picture? How is rainfall related with the monsoon on the Arabian Peninsula. Is chronology such a big problem that the correlation does not work?

References suggested Clark, P. U., Dyke, A. S., Shakun, J. D., Carlson, A. E., Clark, J., Wohlfarth, B., Mitrovica, J. X., Hostetler, S. W., and McCabe, A. M.: The Last Glacial Maximum, Science, 325, 710-714, 2009. Levine, R. C. and Turner, A. G.: Dependence of Indian monsoon rainfall on moisture fluxes across the Arabian Sea and the impact of coupled model sea surface temperature biases, Clim. Dyn., 38, 2167-2190, 2012. Mahesh, B. S. and Banakar, V. K.: Change in the intensity of low-salinity water inflow from the Bay of Bengal into the Eastern Arabian Sea from the Last Glacial Maximum to the Holocene: Implications for monsoon variations, Paleogeogr. Paleoclimatol. Paleoecol.,

397, 31-37, 2014. Vijith, V., Vinayachandran, P. N., Thushara, V., Amol, P., Shankar, D., and Anil, A. C.: Consequences of inhibition of mixed-layer deepening by the West India Coastal Current for winter phytoplankton bloom in the northeastern Arabian Sea, J. Geophys. Res.-Oceans, 121, 6583-6603, 2016.

---

## Author Comment (AC2) · 15 Jan 2018

**Reply to the Referee #2 comments**

The authors are thankful to referee for his/her critical and constructive comments which helped in improvising the manuscript. The necessary changes in view of the comments made by the referee have been incorporated in the manuscript and will be sent after the editor's decision. Here, we provide replies to the comments.

**Comment:** *The comparison of the productivity record with the SST records is misleading. As the authors state, high productivity could be expected during periods of strong upwelling, i.e. low temperatures. The SST records, however, are dominated by the strong glacial-interglacial temperature increase. So during this phase it looks like SST and productivities are positively correlated. The authors discuss this (chapter 4.2.1.) and thus start the comparison with a phase when it does not work; so the Figure still does not help much with the data interpretation. A comparison of other productivity records, concentrating on the Somali and Oman upwelling areas could be more illustrative. The SST records of the Holocene (after the strong glacial-interglacial) may be plotted with reversed scale in order to better illustrate whether and when there is an anti-correlation. In addition: there is some discussion in Huguet et al. (2006) about the TEX86 temperatures; it may not represent annual average temperatures but has a SW monsoon bias. This needs to be addressed and could actually support the authors.*

*Reply:* The high productivity is expected during the periods of strong upwelling and low SST, but only during southwest monsoon. Previous studies have shown that the southwest monsoon was weak/absent during LGM. Hence, modern relation of productivity and upwelling would not exist during LGM. The Mg/Ca based SST records are dominated by glacial-interglacial signal as it was measured in planktonic foraminifera G.*ruber*, which occurs throughout the year. But biogenic silica flux is dominated by southwest monsoon signal. The biogenic silica flux thus serves as a better proxy for upwelling rather than SST. This major observation is being underscored by comparing biogenic silica flux and SST and has been further elaborated in the revised manuscript. The other productivity records from Somali and Oman upwelling regions which record annual signal are either based on calcareous microfossils or organic matter. Therefore, comparison of biogenic silica flux with other productivity records would be improper. The TEX86 proxy related points of Huguet et al., (2006) have been included in the revised manuscript.

**Comment:** *The comparison of the biogenic opal fluxes form the Somali upwelling and the delta18O (precipitation) records is a new idea (not published by Tiwari et al.) but is too vague to be the main part of the paper. The anti-correlation of Western Ghats precipitation with western Arabian Sea upwelling was modelled for the present Arabian Sea and the authors cite only one paper (Izuma, see above). As the authors also discuss, differences in evaporation and also surface water inflow from the Bay of Bengal have impacted the salinity off the west coast of India during the past so that much of the changes are related to several different processes (see Vijit et al., 2016; Mahesh and Banakar, 2014). Furthermore, even the present relationship between precipitation on the Indian Subcontinent and upwelling/productivity in the Arabian Sea is not very clear (see Levine and Turner, 2012) so this topic needs at least some further discussion.*

**Reply:** Though the anti-correlation between western Arabian Sea upwelling/SST and rainfall in southwestern India is still open for investigation but few studies show evidence (Shukla, 1975; Arpe et al., 1998; Vecchi and Harrison, 2004; Izumo et al., 2008; Gimeno et al., 2010; Levine and Turner, 2012). Based on these modern observations, the comparison of upwelling and rainfall on longer time scales becomes an important aspect towards its understanding. As per reviewer's suggestion more studies on the modern climate are included in the revised manuscript.

**Comment:** *In the paper by Tiwari et al. (2010), which the authors cite, more data on core SS4018 are available such as carbonate contents and stable isotopes of carbon and nitrogen. These data can be utilized to better understand the processes in the Somali upwelling and would help to better understand the Holocene productivity changes. Tiwari et al. come to similar conclusions, e.g. that productivity does not decline during the late Holocene despite the decreasing insolation, based on a multiproxy study. The authors have now additional evidence that this is the case and can prove what Tiwari et al. suggested: the decline of carbonate could be due to the replacement of carbonaceous by siliceous primary producers. The carbonate/opal ratio could show this and strengthen the authors' point. The published data need to be included and elaborated on.*

**Reply:** Yes, there are more proxy data available in the same core. We also agree that Tiwari et al., 2010 have suggested siliceous productivity as an alternative for calcareous. The whole core is composed of carbonaceous sediments with some minor variations. The carbonate content variations in this case is a function of nutrient availability and upwelling due to southwest monsoon variability. Carbonaceous productivity requires nutrients and micro nutrients. Oceanic regions (like Southern Ocean) deficient in micro nutrients, equatorial regions and high upwelling regions are known to experience high siliceous productivity (Lizitzin, 1971). Somalia basin

known for strong upwelling, receives excessive nutrients brought from the sub-surface waters, is one region which results in relatively high primary productivity as a function of upwelling (Burkill et al., 1993). Synchronous increase of B.Si/carbonate and biogenic silica flux (Figure r1) attests to increasing upwelling in the Somalia region. However detailed discussion on the findings of Tiwari et al., 2010 have been included in the revised manuscript, as per reviewer's suggestion. If the referee still suggests the inclusion of previously published data by Tiwari et al., 2010 is necessary, then we are ready to incorporate in the revised manuscript.

**Minor Comments:**

**Comment:** *The authors use the term "glacial" and "deglaciation" without giving references for these phases. They should also give the correct time for the beginning of the Holocene. I think that the use of LGP is rather uncommon but LGM is more common and can be referenced (Clark et al, 2009).*

*Reply:* The time for beginning of Holocene is modified from 11 ka to 11.7 ka BP in the revised manuscript text and figures. We agree with the referee that the use of LGM is more common, however, in this paper LGP has been used as it is a cumulative period comprising Heinrich event 1 and part of LGM, also explained in the beginning of chapter 4.2.1.

**Comment:** *Lines 55-59: it does not become clear why biogenic silica appears after carbonate, clarify in more detail.*

*Reply:* Biogenic silica productivity as mentioned earlier are typical for regions like Southern Ocean, Equatorial Regions and high upwelling regions (like the present study site Somalia Basin). Silicate is low as compared to nitrate in surface and the intermediate ocean. Generally, in normal conditions in presence of nutrients and micro nutrients (Fe etc mostly supplied by terrestrial input or aeolian dust) or during the initial phase of upwelling (which brings high nitrate and low silicate water), it is mostly carbonaceous productivity which is dominated. But during high upwelling periods, due to excessive pumping of nutrients (silicate) to surface ocean by sub-surface waters (Haake et al., 1993), after initial carbonaceous productivity, depletion of micro nutrients to sustain excessive nutrient utilization and presence of more silicate results in siliceous productivity.

**Comment:** *Line 68: Is the age model used here different from the one used by Tiwari et al. for the same core, if yes, why? Is the same rate of sedimentation used for the whole core, despite available C14 ages? Why?*

*Reply:* Yes, we have used constant sedimentation rate to compute the flux. We considered that the variation in the sedimentation rate between 3-23 cm.ka-1 in our 4018 core as published by Tiwari et al., (2010), is a result of age control point selection. To minimize this effect, we computed an average sedimentation rate for the whole core.

**Comment:** *Lines 130-134: difficult to understand, explain in more detail. Why should variations be three times greater?*

*Reply:* If the observed variation in the biogenic silica flux is dominated by changes in burial efficiency (BE), low BE can be attributed to low flux and high BE to high flux i.e. result of low flux divided by low BF should be equal to high flux divided by high BE. High and low BE were assigned as per modern observation by Konning et al., 2001. In the present study, it was observed that the ratio of flux to BE was three times greater at the top as compared to the bottom, indicating the absence of preservation effect and the change in silica flux is exclusively a function of upwelling. Detailed explanation is presented in the revised manuscript.

**Comment:** *Lines 145-148: very short and therefore difficult to understand, explain in more detail (see also general comment on the comparison of SST and productivity records above). When do you expect a correlation, when an anti-correlation, why? This cannot be explained in two sentences.*

*Reply:* During southwest monsoon, with increase in upwelling anti-correlation exist between biogenic silica flux and SST. However, there is no relation between biogenic silica flux and SST in absence of southwest monsoon during LGP (18.5-15 ka BP). Detailed discussion on the comparison between biogenic silica flux and SST is now included in the revised manuscript.

**Comment:** *Line 181: I find the use of the deglacial period (DP; 15-11 ka BP) rather problematic as it covers the Pleistocene/Holocene boundary.*

*Reply:* We have used the deglacial period as the connecting phase between Holocene and LGP. The time range for deglacial period has been modified as 15-11.7 ka BP in the revised manuscript.

**Comment:** *Line 185: what does "entrainment of the SW monsoon" mean?*

*Reply:* We want to state that the northern limit of southwest monsoon was attained at the beginning of deglacial period onto the study site. The sentence has been modified as suggested.

**Comment:** *Lines 197-204: these lines again show that the comparison of moisture and upwelling does not work (see above). So when does it work and is it at all useful to show it for the whole period?*

*Reply:* The change in the relationship between upwelling and rainfall (moisture) at ~11 ka BP is the major finding of the present study. The upwelling-rainfall interaction was different during deglacial period than Holocene as well as modern. So the comparison for the whole period is necessary.

**Comment:** *Lines 221: very short and not convincing. How does the record from the Qunf Cave come into the picture? How is rainfall related with the monsoon on the Arabian Peninsula? Is chronology such a big problem that the correlation does not work?*

*Reply:* This part has been elaborated in the revised manuscript. Fleitmann et al., 2007 have used Qunf cave record as indicator of southwest monsoon rainfall. The location of the Qunf speleothem is very close to the study area. If southwest monsoon was the reason for rainfall in southern Oman then western Arabian Sea must be the moisture source and is the basis for comparing upwelling and Qunf cave record. This aspect is now included in the revised manuscript. Chronology limits the comparison of short time variations in upwelling and rainfall records, however, long-term trend comparison is possible.

**References suggested:**
1. Clark, P. U., Dyke, A. S., Shakun, J. D., Carlson, A. E., Clark, J., Wohlfarth, B., Mitrovica, J. X., Hostetler, S. W., and McCabe, A. M.: The Last Glacial Maximum, Science, 325, 710-714, 2009.
2. Levine, R. C. and Turner, A. G.: Dependence of Indian monsoon rainfall on moisture fluxes across the Arabian Sea and the impact of coupled model sea surface temperature biases, Climate Dynamics, 38, 2167-2190, 2012.
3. Mahesh, B. S. and Banakar, V. K.: Change in the intensity of low-salinity water inflow from the Bay of Bengal into the Eastern Arabian Sea from the Last Glacial Maximum to the Holocene: Implications for monsoon variations, Paleogeogr. Paleoclimatol. Paleoecol., 397, 31-37, 2014.
4. Vijith, V., Vinayachandran, P. N., Thushara, V., Amol, P., Shankar, D., and Anil, A. C.: Consequences of inhibition of mixed-layer deepening by the West India Coastal Current for winter phytoplankton bloom in the northeastern Arabian Sea, J. Geophys. Res.-Oceans, 121, 6583-6603, 2016.

*Reply:* Suggested studies are included in the revised manuscript.

---

## Author Response (AR2)

**Reply to the Referees' comments**

The authors are thankful to all the three referees for their critical and constructive comments which helped in improving the manuscript. The necessary changes in view of the comments made by the referee have been incorporated in the manuscript. Here, we provide replies to the comments. In view of the comments and suggestions made by the reviewers, we have changed the title of the manuscript from "Strengthening of the Somali upwelling during the Holocene and its impact on southwest monsoon rainfall" to "Variations of the Somali upwelling since 18.5 ka BP and its relationship with southwest monsoon rainfall". The manuscript presently focuses on variation of the biogenic silica during the last 18.5 ka and possible implications.

**Referee #1**

We thank the referee for his recommendation and positive view on our manuscript.

**Referee #2**

**Comment:** *Line 11: 18.5-15 ka is referred to as the glacial maximum; Line 12: the BA is referred to as post-glacial. The latter term is misleading as the Holocene starts at 11.7 ka.*

*Reply:* The Bølling-Allerød (15-12.9 ka BP) is generally considered to be a warming phase after the last glacial maximum. We did not refer to the B/A event as post glacial one, but the upwelling signal shows an increase during the B/A. This increase in upwelling signal after last glacial maximum is referred here as post glacial intensification of Southwest monsoon. The sentence has been rephrased in the revised manuscript.

**Comment:** *Lines 38-39: the introduction to biogenic silica is too short here. But 4.1 would fit in here.*

*Reply:* The introduction to biogenic silica has been modified by bringing the discussion part as per reviewer suggestion.

**Comment:** *Lines 75ff: repeated in the end of chapter 4.1; see comments on lines 156-162*

*Reply:* The repeated sentences were deleted in the section 4.1 (now part of introduction) from line 158 to 160 (now 80 to 83).

**Comment:** *The first chapter (4.1) is a very useful part of an introduction because it outlines the problem of silica preservation and thus the suitability of this proxy as a productivity indicator. As own data are not discussed this is certainly not part of a discussion.*

**Reply:** We agree with the referee that the chapter 4.1 (now part of introduction) does not contain own data, however it contains the synthesis of the previous study in the study area and gives a complete idea about the proxy. As advised, this chapter has been moved to introduction to give a more comprehensive details on biogenic silica.

**Comment:** *The last lines 156-162 of chapter 4.1 repeat what has already been written in the methods. However, I find this part still very problematic and I am still not sure if I understand it correctly. I guess that the authors used the age model of Tiwari et al. (2010) which is of course what should be done. Second, I understand that the authors used an average sedimentation rate to calculate the biogenic opal accumulation rates. This cannot be done! As Fig. 4 shows and as the authors also state there are large variations of the sedimentation rates with lower rates during the Holocene and higher during the glacial section. If an average rate of the entire core is used these differences are concealed. I still don´t understand why this is done and I think that it is not justified to do it. The authors could compare % biogenic Si with accumulation rates and discuss whether the differences are related to winnowing/focusing, they could also discuss problems with accumulation rates but they cannot blur these differences by using an average sedimentation rate.*

**Reply:** The repeated lines in the chapter 4.1 (now in introduction) have been deleted as per reviewer's suggestion. The average sedimentation rate has been used to overcome the problems of sediment redistribution if at all present in the study area. However, we had no intentions to hide the use of average sedimentation rate and thus have shown the age-depth model in figure 4 and discussed about the validity of its use in the present study (lines 125-127). This is based on the similar method used in a referred study by deMenocal et al.,2000 in east Atlantic sediment core to understand the west African climate.

**Comment:** *The discussion of the applicability of different temperature proxies (4.2.) is certainly dispensable in this paper as it does not produce own SST data. At least until line 185 the whole section could be deleted.*

**Reply:** The chapter 4.2 (now 3.1) was added during the first revision of this manuscript as per reviewers' comment on the comparison of biogenic silica flux and SST. We agree with the reviewer that the introduction of different SST proxies in chapter 4.2 is very extensive. Major part of chapter 4.2 have been now deleted in the corrected manuscript as suggested.

**Comment:** *The next extensive discussion of published data (4.3, lines 198-234) is more important to the data interpretation as the authors discuss how Arabian Sea SST (modulated by upwelling) and moisture transport to the Indian Subcontinent are related. This discussion would need to be focused and shortened. From line 229 to 234 it is much better.*

**Reply:** We have tried to discuss this at length given the complex nature of relation between the Somali upwelling and Indian rainfall and also to give a fair view on the uncertainties around this relationship.

**Comment:** *Results and discussion are mixed and the points are not very clearly presented. The results section could therefore be deleted.*

**Reply:** Result section has been deleted in the revised manuscript as per referee's suggestion.

**Referee #3**

**Comment:** *The major part of the paper and discussion (Line 198 - 273) deals with the time period between 18.5 to 11.7 ka BP including Last Glacial Period (18.5 to 15 ka BP) and Deglacial period (15-11.7 ka BP), while only one portions (Line 275 to 310) deals with the Holocene. But paper title gives an impression that entire work is on Holocene time period. I will suggest to make changes in the title accordingly like "Variations of the Somali upwelling since 18.5 ka BP and its relationship with southwest monsoon rainfall".*

**Reply:** We accept the reviewer's concern and changed the manuscript title to "Variations of the Somali upwelling since 18.5 ka BP and its relationship with southwest monsoon rainfall", as per the suggestion.

**Comment:** *It is still not clear that, how average sedimentation rate is different from the sedimentation rate given in the Tiwari et al., 2010.*

**Reply:** Age-depth model presented in Tiwari et al., 2010 was based on the age control points shown in figure 4 and their sedimentation rate was calculated using the variation in two adjacent age control points. However the average sedimentation rate method considers only two dates (minimum and maximum) in the sediment core and give a constant sedimentation rate for the whole core, which is mainly used to overcome the problems associated with sediment redistribution.

**Comment:** *In Figure 5, 6, 7 the Biogenic silica flux shows increasing trend from ~17 to 13.5, ~11.7 to 9.7, ~8.2 to 5, and ~4 to 2 ka BP and sharp decrease between ~13.5 and 11.7, ~9.7 and 8.2 and ~5 and 4 ka BP. What processes is attributed to sharp decrease in upwelling*

*during these periods like 13.5 to 11.7 attributed to Younger Dryas. The Mg/Ca SST is annual SST measured by Anand et al., 2008, while anti-phase relationship between SST and upwelling is seasonal phenomenon (Fig.3) remain for a quarter of year as suggested by the authors. It means Mg/Ca SST shows major glacial/interglacial variations in the SST, than, why authors are trying to co-relate with the upwelling proxy?*

*Reply:* The short term variations in the biogenic silica flux were discussed in the chapter 3.2. We have discussed the possible reasons for the variation in the flux. SST is physically connected with upwelling, and an important proxy for reconstruction of upwelling history. The figure 1 (now figure 1) shows the modern data set with the SST values being measured directly. But the Mg/Ca proxy is based on the foraminifera abundance which may or may not preserve the seasonal SST signal based on the productivity dynamics in a region. For example, if the foram abundance was influenced by seasonal changes in the hydrography then the Mg/Ca of foram would give seasonal SST rather than annual. The reason behind the comparison of biogenic silica with SST, was to show the suitability of biogenic silica flux over SST as upwelling proxy in our study area.

**Comment:** *Various marine as well as land based proxies suggests huge variations in southwest monsoon during the Holocene. The overall biogenic flux remains between 6 to 4 g.m-2.y-1. On close look (Figure 10), the author may observe low southwest monsoon during low upwelling (i.e., 4 g.m-2.y-1). Hence, line 312 Somali upwelling possibly had a negative impact on southwest monsoon rainfall throughout the Holocene is over exaggeration as authors have not done one to one correlation between upwelling and southwest monsoon record.*

*Reply:* We agree with the referee that this study does not present one to one correlation between the Somali upwelling and southwest monsoon rainfall. The overall trends have been therefore compared and interpreted accordingly. Though there are points of anti-correlation, the trend shows a negative impact of Somali upwelling on southwest monsoon rainfall baring the beginning of Holocene. The line 368 has been modified as "It is observed that baring the beginning, the Somali upwelling had a negative impact on southwest monsoon rainfall over south-western India and Oman during Holocene".

**Comment:** *Line 124 – 126 is repetition of line 107 -117*
*Reply:* Result section has been deleted in the revised manuscript.

**Comment:** *Degree symbol is missing at line 178.*
*Reply:* This part has also been deleted in the revised manuscript as per referee #2 suggestions.

[revised manuscript text omitted]

---

## Author Response (AR3)

**Reply to the Editor's comments**

The authors are thankful to the editor for critical and constructive comments which helped in improving the manuscript. The necessary changes in view of the comments made by the editor have been incorporated in the manuscript. The manuscript has been revised for English grammar and structural corrections with help of professional English scientific writer. Here, we provide replies to the comments

**Comment:** *In the abstract you mention ' upwelling during the last glacial period (18.5-15 ka BP) and strengthened upwelling during the Bølling-Allerød (15-12.9 ka BP). Although later in the text you explain the term Last Glacial period in the context of Somali upwelling, a reader will be strongly surprised that the Last Glacial period is defined as 18.5-15 ka BP. I would recommend not to mention last Glacial Period it is clearly defined in the text.*

*Reply:* The term "last glacial period" has been removed from the abstract in the revised manuscript.

**Comment:** *Line 39- Modify to its chemistry*

*Reply:* The text has been appropriately modified.

**Comment:** *Line 42- replace "has" with "have"*

*Reply:* The changes have been incorporated in the revised manuscript.

**Comment:** *In the introduction (page 3, line 64) you mention ' in the present record'. However, 'this record' has not been previously mentioned in the text. It is only later mentioned in the last paragraph of the introduction, when the aims of the study are summarized. My suggestion is to move this last paragraph ('The present study thus aims..') ti line 44 in the introduction, just before the paragraph starting with 'The surface waters...'). It is perhaps not an ideal solution, but it would avoid more drastic reorganizations of the introduction. Ideally, it would be better to state the aims of the study earlier in the introduction, possibly in the first or second paragraph, but I have not identified a suitable place without further extensive re-phrasing.*

*Reply:* The last paragraph of the introduction is moved to line number 45 in the revised manuscript.

**Comment:** *Line 50- replace "under saturated" with "undersaturated".*

*Reply:* The change has been incorporated in the revised manuscript.

**Comment:** *Line 95- Delete comma.*

*Reply:* The change has been incorporated in the revised manuscript.

**Comment:** *Line 97- rephrase the sentence using "more nitrate and phosphate than silicate reach the surface".*

*Reply:* The change has been incorporated in the revised manuscript.

**Comment:** *Line 101- replace "to" with "that".*

*Reply:* The change has been incorporated in the revised manuscript.

**Comment:** *Line 103- replace "and" with "was".*

*Reply:* The change has been incorporated in the revised manuscript.

**Comment:** *Line 140- Do not capitalize the first letter in Aluminium and Silicon.*

*Reply:* The change has been incorporated in the revised manuscript.

**Comment:** *Line 155- replace the sentence "While, the biogenic silica flux has been controlled by the SWM upwelling signal since it is produced during the southwest monsoon season and preserves upwelling signal" with "However, the biogenic silica flux is controlled by the SWM upwelling after its production during the southwest Monsoon season, and therefore preserves the upwelling signal".*

*Reply:* The change has been incorporated in the revised manuscript.

**Comment:** *Line 159- Add the word "last" before 18.5 ka. .*

*Reply:* The word "last" has been added at the suggested place in the revised manuscript.

**Comment:** *Line 162- replace "on temporal scale" with "over time".*

*Reply:* The change has been incorporated in the revised manuscript.

**Comment:** *Line 163- delete "is" after TEX86 SST and add "being" after "always".*

*Reply:* The change has been incorporated in the revised manuscript.

**Comment:** *Line 165- However, during the last 11.7 ka the Mg/Ca based SST shows a strong anti-correlation with biogenic silica flux record, indicating variation in of influence of on the seasonal signal for of different proxies (Fig. 8). Modify sentence using "indicating variation in the influence of the seasonal signal". However, I cannot see how this firm conclusion can be derived from the strong anticorrelation.*

*Reply:* The sentence has been modified in the revised manuscript.

Biogenic silica flux is controlled by upwelling during the southwest monsoon season, thereby preserves the seasonal signal. If a SST records the southwest monsoon seasonal signal, it should show an anti-correlation with biogenic silica flux as upwelling increases biogenic silica flux and decreases SST. However, in the present study the anti-correlation is strong between biogenic silica flux and TEX$_{86}$ SST during 15 to 11.7 ka BP, but Mg/Ca SST shows a strong anti-correlation with biogenic silica flux record during the last 11.7 ka. The change in relation between biogenic silica flux and each SST record indicates variation in the ability of each SST proxy to record seasonal signal.

**Comment:** *Line 180- capitalize first letter in "pacific".*

*Reply:* The change has been incorporated in the revised manuscript.

**Comment:** *Line 188- replace "Though" with "However".*

*Reply:* The change has been incorporated in the revised manuscript.

**Comment:** *Line 190- Sea SST during SWM) and SWM rainfall. Both Somali upwelling and SWM rainfall being caused by southwest monsoonal winds during SWM season, hence their anti-correlation indicates a negative feed-back within the system.*

*I do not think that negative feedback is the right word here. Negative feedback would indicate that a given variable reacts to oppose the initial perturbation. Here, the text just shows there is an anticorrelation between upwelling and rainfall, but there is not reaction of any of the variables to oppose the influence of a driver. Also, the word 'system' is ambiguous. Which system ? it has not been defined before.*

*Reply:* We agree with the editor that the use of the terms "feedback" and "system" are inappropriate in the noted sentence and deleted in the revised manuscript.

**Comment:** *Line 204- replace "Though, there are no observational study" with "However, there is no observation studies".*

*Reply:* The change has been incorporated in the revised manuscript.

**Comment:** *Line 236- Delete comma after "rainfall".*

*Reply:* The change has been incorporated in the revised manuscript.

**Comment:** *Line 266- replace "expanse" with "extension".*

*Reply:* The change has been incorporated in the revised manuscript.

**Comment:** *Line 284- last 8 ka suggesting reduction in SWM rainfall (Fig. 10b). The hiatus in Oman speleothem record at 2 ka BP coincides with the strengthened Somali upwelling, however, is difficult to explain (Fig. 10a & b). The SK-17 record (Anand et al., 2008).*

*However, it is difficult to explain. Please be more specific. What is difficult to explain ?*

**Reply:** The sentence has been modified to "*The hiatus in Oman speleothem record at 2 ka BP coincides with the strengthened Somali upwelling. Strengthened Somali upwelling at 2 ka BP might have reduced the moisture supply for the SWM rainfall over Oman and caused hiatus in the speleothem record*" in the revised manuscript.

**Comment:** *The last two conclusions are very speculative, and I do not think they can be derived from the results. The result section itself indicates that a change in the source of moisture could explain the change in the correlation between upwelling and rainfall, but this is just one plausible explanation. I do not think it can be listed as one firm conclusion of the study. Also, the last conclusion is rather a recommendation for future studies, but at least for modelling studies it is not really useful. Climate models simulate by themselves any variations in the precipitation source. Modelers do not need to incorporate them ad-hoc.*

**Reply:** The last two points in the conclusion chapter were removed in the revised manuscript.

.

[revised manuscript text omitted]